# BMW: Bidirectionally Memory bank reWriting for Unsupervised Person Re-Identification

**Xiaobin Liu**
Nankai University
Tianjin, China
`liuxb@nankai.edu.cn`

**Jianing Li**[*]
Hong Kong Polytechnic University
Hong Kong, China
`tensor.li@polyu.edu.hk`

**Baiwei Guo**
Huawei Technologies Ltd.
Shanghai, China
`baiwei.guo@epfl.ch`

**Wenbin Zhu**
Nankai University
Tianjin, China
`wenbinzhu@mail.nankai.edu.cn`

**Jing Yuan**[*]
Nankai University
Tianjin, China
`nkyuanjing@gmail.com`

## Abstract

Recent works show that contrastive learning based on memory banks is an effective framework for unsupervised person Re-IDentification (ReID). In existing methods, memory banks are typically initialized with cluster centroids and rewritten with positive samples via the momentum mechanism along with the model training. However, this mechanism solely focuses on the intra-class compactness by pulling memory banks close to positive samples, neglecting the inter-class separability among different memory banks. Rewriting memory banks with partial constraint limits their discrimination capacities, and hence hinders learning discriminative features based on those memory banks. In this paper, we claim that memory banks should be rewritten with both intra-class and inter-class constraints, and therefore propose a unified memory bank rewriting mechanism, Bidirectionally Memory bank reWriting (BMW), to chase enhanced discrimination capacity. Specifically, BMW formulates the memory bank rewriting as the gradient descent update with two objectives, *i.e.*, reducing intra-class diversity and enhancing inter-class separability. To effectively enhance the separability of memory banks with limited number of rewriting steps, we further design a novel objective formulation for the inter-class constraint, which is more effective for one step update. BMW enhances both representation and discrimination capacities of memory banks, thus leads to an effective ReID feature optimization. BMW is simple yet effective and can serve as a new paradigm for person ReID methods based on memory banks. Extensive experiments on standard benchmarks demonstrate the effectiveness of our BMW method in unsupervised ReID model training. Specially, BMW even outperforms previous methods that use stronger backbones. Code is available at https://github.com/liu-xb/BMW.

## 1   Introduction

Person Re-Identification (ReID) aims to match a query person image in a gallery set, making it an important task in intelligent surveillance systems [1–4]. Currently, supervised person ReID methods have been widely studied from different aspects and achieved promising performance [5–14]. However, supervised methods requires laborious manual person ID annotation, limiting their applications in real-world scenarios lacking annotated data. To address this issue, recent researchers

---

[*]Corresponding Author

39th Conference on Neural Information Processing Systems (NeurIPS 2025).

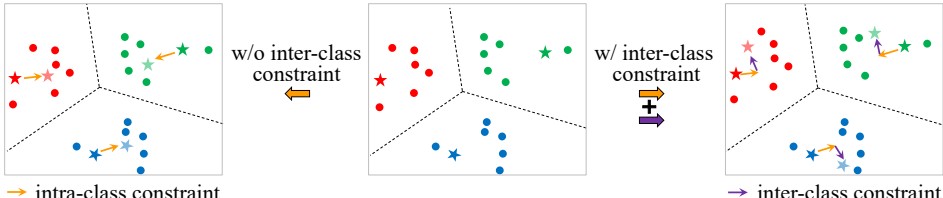

Figure 1: Illustration of the memory bank update without (w/o) and with (w/) the inter-class constraint. Dots and stars denote features and memory banks. Different colors denote different clusters. Without the inter-class constraint, memory banks are pulled to approach positive samples only (shown as yellow arrows). With the inter-class constraint, memory banks are also push away from each other (shown as purple arrows), thus enhancing their separability.

focus on unsupervised person ReID, and multiple methods are proposed from different aspects, such as pseudo label prediction [15, 16], unsupervised metric learning [17, 18], and image synthesis [19, 20]. More details of related works are summarized in Sec. 2.

Currently, state-of-the-art unsupervised person ReID methods adopt the contrastive learning framework based on memory banks [17, 21–26]. These methods typically use the centroid of each cluster obtained by unsupervised clustering methods to initialize the memory bank. During model training, those methods rewrite memory banks with training samples of corresponding clusters by the momentum mechanism, *i.e.*, rewritten memory banks are weighted summation of the initial ones and positive samples. Features are optimized by contrastive loss to maximize the similarities with corresponding memory banks and minimize the similarities with others.

Despite the significant success of those methods, there still remain two open issues unexplored. Firstly, as memory banks serve as the anchors for the feature optimization, they are required to be both intra-class representative and inter-class separative to supervise the discrimination capacities of features. However, the previously used momentum mechanism for memory bank rewriting solely focuses on learning representative memory banks for each class, neglecting the inter-class constraint on memory banks. This limits the separability inside memory banks, hence hinders effective feature optimization base on those memory banks. Secondly, compared with model parameters that are entirely updated in each iteration, only several memory banks corresponding to selected clusters are rewritten once in each iteration. This requires an effective objective formulation to enhance the separability with few rewriting steps. Existing metric learning methods commonly formulate the inter-class separability constraint as enlarging the Euclidean distance. However, this always leads to larger feature norms, which is ineffective for memory bank rewriting after L2 normalization.

This paper is motivated to study an effective memory bank rewriting mechanism for the unsupervised person ReID model training. Different from existing methods, we innovatively formulate the memory bank rewriting as the gradient decent update for the objective function. From this respective, the previously used momentum mechanism corresponds to the objective function containing only the intra-class constraint for the representability, thus limits the separability of memory banks. To handle this issue, this paper complement the objective function with inter-class constraint to additionally ensure the discrimination capacity of memory banks. As shown in Fig. 1, the extra inter-class constraint enhances the separability of memory banks, thus it further enables effective feature optimization. We further propose an effective objective formulation for the inter-class constraint by pulling a memory bank close to the opposite vector of memory banks for other classes. We show that this is more effective in enhancing separability for one step update. A dynamically weighting strategy is also designed to ensure the awareness of importance of different constraint component.

Our method jointly rewrite memory banks with the above two constraints. We hence call our method as Bidirectionally Memory bank reWriting (BMW). We test our BMW on standard benchmarks including Market-1501 and MSMT17. Comparison with recent works shows that our BMW is able to enhance the discrimination capacity of features learned with memory bank. For instance, BMW boosts the mAP accuracy from 81.2% to 86.3% on Market-1501, outperforming existing both unsupervised and transfer learning methods by a clear margin. Moreover, BMW even outperforms recent DCC [23] which uses stronger backbone, *i.e.*, 86.2% by BMW with ResNet50 *v.s.* 85.8% by DCC with ResNet50-ibn on Market-1501.

The contributions of this paper can be summarized as follows:

1) To the best of our knowledge, this is the first effort to analyse the memory bank rewriting mechanism from the perspective of gradient decent update, which could provide certain insight for future studies on methods based on memory banks.

2) For updating memory banks by the gradient decent strategy, we proposes the Bidirectionally Memory bank reWriting (BMW) method which involves both intra-class and inter-class constraints in the objective formulation. And a new objective formulation for the inter-class constraint is proposed to ensure effective update of memory banks in limited steps.

3) BMW enhances both the representation and discrimination capacities of memory banks and thus can boost the feature optimization. Experimental results with superior performance against state-of-the-art methods demonstrate the effectiveness of BMW.

## 2  Related Work

**Domain adaptive person ReID** uses both a labeled source dataset and the unlabeled target dataset for model training and aims at transferring knowledge contained in labeled data to the target data. Some works targeting to narrow the domain gap between the source data and the target data in image space by GANs [27, 28, 19, 29, 30, 20, 31]. For example, Liu *et al.* [30] propose an adaptive transfer network to effectively transfer images from label domain to unlabeled domain. Several works try to bridge domain gaps by mapping labeled and unlabeled images to a shared feature space [32, 33]. However, these methods always require labeled source data for training, which is hardly available due to privacy issue in real-world applications. Compared with them, this paper only uses unlabeled target data, thus is more suitable for real-world applications.

**Unsupervised person ReID** only requires the unlabeled target data for training and has been studied from two main perspectives, *i.e.*, pseudo label prediction and unsupervised optimization. Some works locally predict pseudo labels for unlabeled person images [21, 34]. For example, Yu *et al.* [34] select positive samples in training batches. However, local label prediction is not precise and will mislead the distance optimization. To acquire reliable pairs, some works have designed unsupervised clustering methods [15, 16] and re-rank method [35]. Based on assigned labels, researchers propose multiple metric learning methods for the unsupervised model training [36, 37, 34, 18, 38, 39, 21]. For example, Liu *et al.* [37] propose a graph consistency constraint between student models and momentum average models for unsupervised model training. Some works adopt additional cues to boost performance [40–45]. For examples, Fu *et al.* [43], Zhang *et al.* [44] and Cho *et al.* [45] use local features to boost the performance. While, experiments show that our model outperforms these methods by a clear margin even without additional cues.

**Memory bank based methods**  are previously proposed for unsupervised feature learning [46–48]. They use memory banks to store features of all features or selected features for the contrastive loss computation. Currently, memory bank based methods dominate the unsupervised person ReID task and achieve promising performance [21, 25, 18, 17, 22, 49, 50, 23, 51, 24]. For examples, Zhong *et al.* [21, 25], Liu *et al.* [18] and Wang *et al.* [52] store each image features in a memory bank for metric learning. Dai *et al.* uses store centroids of clusters in the memory bank, which is followed by many works [23, 35]. Some researchers propose to store different features in memory banks [17, 49, 51, 24]. For instance, Ge *et al.* [17] propose a hybrid memory bank that sore both centroids and outliers. However, these methods only use intra-class positive training samples to update corresponding memory banks. They neglect the inter-class constraint among different memory banks, limiting the separability of them and hindering effective feature optimization.

Compared with existing memory bank based methods, this paper proposes a innovative perspective for memory bank rewriting, *i.e.*, the gradient descent update for the objective function. Guided by this perspective, we propose to combine both the intra-class and inter-class constraints in the objective function, *i.e.*, Bidirectionally Memory bank reWriting (BMW). Experiments show that the proposed BMW outperforms existing methods by a clear margin.

## 3  Overview

Given an unlabeled dataset $\mathcal{D}$, unsupervised person ReID aims to learn the ReID model on $\mathcal{D}$ without annotation. $\mathcal{D}$ can be denoted as $\{x_i | i = 1...N\}$. $x_i$ and $N$ denote the $i$-th image and the number

of images in $\mathcal{D}$, respectively. We denote the feature extraction as $f_i = \Phi(x_i, \theta)$. $\Phi$ and $\theta$ denote the CNN feature extractor and its parameters, respectively.

We adopt the contrastive learning framework based on memory bank for the unsupervised training on $\mathcal{D}$. Pseudo ID labels on $\mathcal{D}$ can be generated by unsupervised clustering methods before model training. After clustering features into $C$ clusters, a centroid feature for each cluster is computed by averaging all features in this cluster, resulting in $C$ centroids corresponding to $C$ clusters. These centroids for each cluster are used to initialize a memory bank $\mathcal{M}$, $i.e.$, $\mathcal{M}[c]$ is initialized by the centroid feature of the $c$-th cluster with L2 normalization.

Via treating $\mathcal{M}$ as a non-parametric classifier, the contrastive loss is used for model training following previous works [23, 22, 35]. For $x_i$ belonging to the $c$-th cluster, the contrastive loss is computed as:

$$\mathcal{L}_\theta(x_i) = -\log \frac{exp(f_i \cdot \mathcal{M}[c]/\tau)}{\sum_{j=1}^{C} exp(f_i \cdot \mathcal{M}[j]/\tau)}, \tag{1}$$

where $\tau$ is temperature hyper-parameter to adjust the scale of similarity distribution. Along with the model training, $\mathcal{M}$ is rewriting based on both training samples and itself by the proposed BMW method, which will be given in Sec. 4.

## 4 Bidirectionally Memroy bank reWriting

Existing methods rewrite $\mathcal{M}$ with positive training samples via the momentum mechanism. For example, it rewrites $\mathcal{M}[c]$ with $f_i$ belonging to the $c$-th cluster can be formulated as:

$$\mathcal{M}[c] \leftarrow \alpha\mathcal{M}[c] + (1-\alpha)f_i, \tag{2}$$

where $\alpha$ is used to adjust the update rate of memory banks.

In this paper, we propose a novel perspective of memory bank rewriting, which formulates the rewriting procedure as the gradient decent update for objective function $\mathcal{L}_\mathcal{M}$ as follows:

$$\mathcal{M}[c] \leftarrow \mathcal{M}[c] - \frac{\partial \mathcal{L}_{\mathcal{M}[c]}}{\partial \mathcal{M}[c]}. \tag{3}$$

Note that the update rate is omitted in Eqn. 3 and will be used for each constraint of $\mathcal{L}_\mathcal{M}$. To enhance both the representation and the discrimination capacities of $\mathcal{M}$, the proposed BMW formulates $\mathcal{L}_\mathcal{M}$ as the combination of both the intra-class constraint $\mathcal{L}_{intra}$ and the inter-class constraint $\mathcal{L}_{inter}$ as:

$$\mathcal{L}_\mathcal{M} = \lambda_{intra}\mathcal{L}_{intra} + \lambda_{inter}\mathcal{L}_{inter}, \tag{4}$$

where $\lambda_{intra}$ and $\lambda_{inter}$ denotes loss weights. Then Eqn. 3 can be rewritten as:

$$\mathcal{M}[c] \leftarrow \mathcal{M}[c] - \lambda_{intra}\frac{\partial \mathcal{L}_{intra}(\mathcal{M}[c])}{\partial \mathcal{M}[c]} - \lambda_{inter}\frac{\partial \mathcal{L}_{inter}(\mathcal{M}[c])}{\partial \mathcal{M}[c]}. \tag{5}$$

To enhance the rewriting efficiency, the hard sampling strategy is used to compute $\mathcal{L}_\mathcal{M}$, $i.e.$, the farthest positive sample in each training batch and the closest negative memory bank vector for $\mathcal{M}[c]$ are used to compute $\mathcal{L}_{intra}$ and $\mathcal{L}_{inter}$, respectively.

### 4.1 Intra-class Constraint

$\mathcal{L}_{intra}$ aims to enhance the representation capacity of $\mathcal{M}$, which can be designed to reduce the distance between $\mathcal{M}$ and corresponding positive sample. Commonly, multiple samples of one selected cluster are used in a training batch and a cluster commonly contains a limited number of samples, $e.g.$, 16 samples of one cluster are sampled in a training batch in previous works [53, 22] and less than 30 samples in each cluster. Thus, $\mathcal{L}_{intra}$ can be designed to reduce the distance between $\mathcal{M}$ and corresponding positive training samples in each iteration. The intra-class constraint between $\mathcal{M}[c]$ and $f_i$ belonging to the $c$-th cluster can be formulated as:

$$\mathcal{L}_{intra}(\mathcal{M}[c], f_i) = \frac{1}{2}||\mathcal{M}[c] - f_i||_2^2. \tag{6}$$

Its gradient w.r.t. $\mathcal{M}[c]$ is computed as:

$$\frac{\partial \mathcal{L}_{intra}(\mathcal{M}[c])}{\partial \mathcal{M}[c]} = \mathcal{M}[c] - f_i. \tag{7}$$

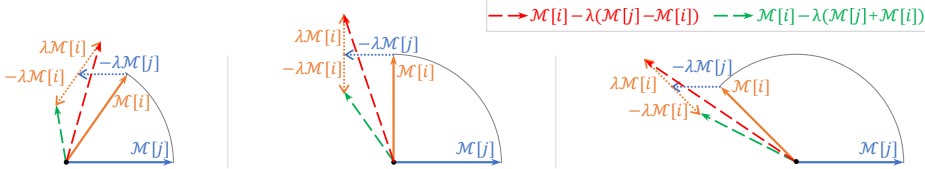

Figure 2: Illustration of the advantage in inter-class constraint of the proposed objective function in Eqn. (8) w.r.t. different angles between $\mathcal{M}[i]$ and $\mathcal{M}[j]$. Yellow arrows and blue arrows denote $\mathcal{M}[i]$ and $\mathcal{M}[j]$, respectively. Green dashed arrows and red dashed arrows denote the updated $\mathcal{M}[i]$ by the proposed objective function in Eqn. (8) and the previous method, respectively. It can be observed that the objective function in Eqn. (8) is more effective in pushing $\mathcal{M}[i]$ away from $\mathcal{M}[j]$.

Memory banks is encouraged to approach positive samples by $\lambda_{intra}$ to acquire the feature updates along with the update of $\theta$.

It is worth noting that setting $\lambda_{intra} = 1 - \alpha$ and $\lambda_{inter} = 0$ makes Eqn. 5 equivalent to the momentum mechanism in Eqn. 2. This indicates that the previously used momentum mechanism is a special case of our BMW that only involves the intra-class constraint.

## 4.2 Inter-Class Constraint

$\mathcal{L}_{inter}$ aims to enhance the discrimination capacity of $\mathcal{M}$. It can be intuitively designed to enlarge the distance between $\mathcal{M}$ and negative samples in $\mathcal{D}$. However, only a small number of clusters are selected in each training batch compared to the total number of clusters, *e.g.*, only 16 clusters are sampled in a single training batch in previous works [53, 22] and there are more 500 clusters in $\mathcal{D}$. Thus, a training batch only contains limited information for the global inter-class constraint. We hence design $\mathcal{L}_{inter}$ to enlarge the distance between different features inside $\mathcal{M}$.

Previous methods for inter-class distance constraint commonly formulate the objective function as the negative value of Euclidean distance between two negative features as: $-\frac{1}{2}\|\mathcal{M}[i] - \mathcal{M}[j]\|_2^2$. Its gradient w.r.t. $\mathcal{M}[i]$ is $\mathcal{M}[j] - \mathcal{M}[i]$. Then, the updated $\mathcal{M}[i]$ with only inter-class constraint is $(1 + \lambda_{inter})\mathcal{M}[i] - \lambda_{inter}\mathcal{M}[j]$, whose norm is larger than 1 for $\lambda_{inter} \in [0, 1]$ [2]. Thus, the distance is enlarged by both enlarging the norm of $\mathcal{M}[i]$ and the angle between $\mathcal{M}[i]$ and $\mathcal{M}[j]$. As $\mathcal{M}[i]$ is L2 normalized after rewriting, only the angle enlarging is retained and the distance enlarging via rewriting will be partially offset.

To chase an effective objective function for the inter-class constraint, we design $\mathcal{L}_{inter}$ to pull $\mathcal{M}[i]$ close to the opposite vector of $\mathcal{M}[j]$, which is formulated as:

$$\mathcal{L}_{inter}(\mathcal{M}[i], \mathcal{M}[j]) = \frac{1}{2}\|\mathcal{M}[i] + \mathcal{M}[j]\|_2^2. \tag{8}$$

Its gradient w.r.t. $\mathcal{M}[i]$ is computed as:

$$\frac{\partial \mathcal{L}_{inter}(\mathcal{M}[i])}{\partial \mathcal{M}[i]} = \mathcal{M}[i] + \mathcal{M}[j]. \tag{9}$$

For the iter-class constraint, rewriting with Eqn. 9 is more effective in enlarging the angle between $\mathcal{M}[i]$ and $\mathcal{M}[j]$, *i.e.*, enlarging the distance between them after L2 normalization. Fig. 2 shows the advantages of the proposed objective function for $\mathcal{L}_{inter}$ in Eqn. 8. We also provide a theory proof of the advantage of Eqn. 8 for one step update in Sec. A.1.

## 4.3 Dynamically Weighting

Directly using the Euclidean distance in Eqn. 6 and Eqn. 8 is not aware of the importance of each pair of samples. This paper proposes to dynamic weight these two constraints based on the idea that larger value of objective function corresponds to higher importance of each component. Hence, we

---

[2] $\|(1 + \lambda_{inter})\mathcal{M}[i] - \lambda_{inter}\mathcal{M}[j]\|_2^2 = 1 + 2\lambda_{inter}(1 + \lambda_{inter})(1 - \mathcal{M}[i]\mathcal{M}[j]) > 1$ as $\mathcal{M}[i]\mathcal{M}[j] < 1$.

weight Eqn. 6 and Eqn. 8 by $\frac{1}{4}||\mathcal{M}[c] - f_i||_2^2$ and $\frac{1}{4}||\mathcal{M}[i] + \mathcal{M}[j]||_2^2$, respectively. Then $\mathcal{L}_{intra}$ and $\mathcal{L}_{inter}$ are re-formulated as:

$$\mathcal{L}_{intra}(\mathcal{M}[c], f_i) = \frac{1}{8}||\mathcal{M}[c] - f_i||_2^4, \quad \mathcal{L}_{inter}(\mathcal{M}[i], \mathcal{M}[j]) = \frac{1}{8}||\mathcal{M}[i] + \mathcal{M}[j]||_2^4. \quad (10)$$

And their gradient w.r.t. $\mathcal{M}[c]$ and $\mathcal{M}[i]$ are computed as (details are provided in A.2):

$$\frac{\partial \mathcal{L}_{intra}(\mathcal{M}[c])}{\partial \mathcal{M}[c]} = (1 - \mathcal{M}[c]f_i)(\mathcal{M}[c] - f_i), \quad \frac{\partial \mathcal{L}_{inter}(\mathcal{M}[i])}{\partial \mathcal{M}[i]} = (1 + \mathcal{M}[i]\mathcal{M}[j])(\mathcal{M}[i] + \mathcal{M}[j]). \quad (11)$$

This weighting strategy focuses on both hard positive pairs, *i.e.*, the pair $f_i$ and $\mathcal{M}[c]$ having small cosine similarity, and hard negative pairs, *i.e.*, the pair $\mathcal{M}[j]$ and $\mathcal{M}[i]$ having large cosine similarity. Eqn. 11 generates larger gradients for hard pairs than easy ones, thus is aware of the pair importance.

## 5 Experiments

### 5.1 Dataset

Experiments are performed on Market-1501 [54] and MSMT17 [27]. Market-1501 contains 32,668 images of 1,501 identities captured from 6 cameras at Tsinghua University. 12,936 images of 751 identities are selected for training, and others are used for testing. In the test set, 3,368 images are selected as query images and remaining 19,732 images are used as gallery images. MSMT17 contains 126,441 images of 4,101 identities captured from 15 cameras at Peking University. 32,621 images of 1,041 identities are selected for training, and others are selected for testing. In the test set, 11,659 images are selected as query images and remaining 82,161 images are used as gallery images. Due to the data usage restriction, experiments on DukeMTMC-reID [55] are provided on the repository on github. Vehicle ReID task [56–59] is similar with the person ReID task. Limited by the paper length, experimental results on the vehicle dataset VeRi-776 [60] are provided in Sec. A.5. Following previous works [38, 23, 35], mAP, Rank1, Rank5 and Rank10 accuracies are reported.

### 5.2 Implementation Detail

The proposed model adopts ResNet50 [61] pre-trained on ImageNet [62] as the feature extractor $\Phi$. The last fully connected layer of ResNet50 is removed and the stride of the last residual block is set to 1. Following previous works [63, 23, 22], a GEM pooling followed by batch normalization layer [64] and L2-normalization layer is added. Experiments with other backbones are provided in Sec. A.3. Parameter $\tau$ in Eqn. (1) is set to 0.05 following [22]. $\lambda_{intra}$, and $\lambda_{inter}$ are set as 0.9, and 0.02, respectively. More details are provided in Sec. A.4.

### 5.3 Model Analysis

In this section, we first analysis the effect of hyper-parameters $\lambda_{intra}$, $\lambda_{intra}$ and $\tau$ on Market-1501 by varying the value of one parameter and keeping others fixed to the optimal value. We then evaluate the proposed inter-class constraint and the dynamically weighting strategy to show their validity. Finally, different sampling methods including hard, average, random and easy are evaluated.

#### 5.3.1 Evaluation on hyper-parameters

$\lambda_{intra}$ affects the update rate for memory banks approaching positive training samples, and thus affects the representation capacity of memory banks. The result of evaluation on $\lambda_{intra}$ is shown in Fig. 3. It can be observed that the performance is insensitive to $\lambda_{intra}$ in an appropriate rage from 0.8 to 0.9. And the best performance is achieved when setting $\lambda_{intra}$ to 0.9. A larger value of $\lambda_{intra}$ gives more emphasis to the intra-class constraint and makes $\mathcal{L}_{inter}$ neglected by $\mathcal{L}_{\mathcal{M}}$. While, a small value of $\lambda_{intra}$ releases the constraint on the representation capacity of memory banks and limits the compactness of intra-class samples.

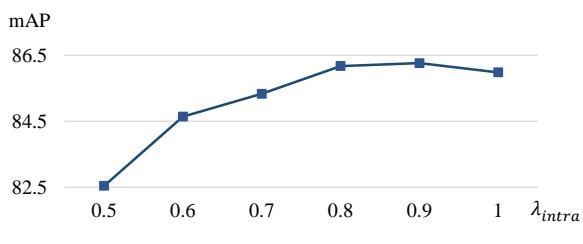

Figure 3: Evaluation on $\lambda_{intra}$.

$\lambda_{inter}$ affects the update rate for pushing memory banks away from each other, and thus affects the discrimination capacity of memory banks. The result of evaluation on $\lambda_{inter}$ is shown in Fig. 4. It can be observed that the best performance is achieved when setting $\lambda_{inter}$ to 0.2. A larger value of $\lambda_{inter}$ gives more emphasis to the inter-class constraint and makes $\mathcal{L}_{intra}$ neglected by $\mathcal{L}_{\mathcal{M}}$. While, a small value of $\lambda_{inter}$ releases the constraint on

Figure 4: Evaluation on $\lambda_{inter}$.

the discrimination capacity of memory banks and thus limits the separability of inter-class samples. Compared with the value of $\lambda_{intra}$, $\lambda_{inter}$ is smaller to achieve the best performance. This is because memory banks are initialized by the centroids of each clusters, thus are already close to positive samples. A larger value of $\lambda_{intra}$ is needed to adapt to the feature update of training samples along with the update of $\theta$. The balance between $\lambda_{intra}$ and $\lambda_{intra}$ is discussed in Sec. A.6.

$\tau$ affects the computation of $\mathcal{L}_{\theta}$ by adjusting the distribution of similarities between training samples and memory banks. The result of evaluation on $\tau$ is shown in Table 1. It is clear that setting $\tau$ to 0.05 achieves the best performance.

Table 1: Evaluation on $\tau$.

| $\tau$ | 0.01 | 0.03 | 0.05 | 0.07 | 0.1 |
|---|---|---|---|---|---|
| mAP(%) | 84.5 | 85.6 | **86.3** | 82.5 | 79.1 |

### 5.3.2 Ablation study

Results of the ablation study on the proposed inter-class constraint and the proposed dynamically weighting strategy is summarized in Table 2. "baseline [22]" denotes the baseline method following the same settings in [22], and neither the inter-class constraint or dynamically weighting strategy is used in the baseline. "w/o $\mathcal{L}_{inter}$" denotes only the intra-class constraint and the dynamically weighting strategy are used. "$||\mathcal{M}[i] - \mathcal{M}[j]||_2^2$ in Eqn. 8" denotes using the Euclidean distance between two memory banks for the inter-constraint objective function as in previous methods.

Table 2: Ablation study of BMW.

| Method | mAP | Rank1 | Rank5 | Rank10 |
|---|---|---|---|---|
| baseline [22] | 79.7 | 90.5 | 96.2 | 97.3 |
| w/o $\mathcal{L}_{inter}$ | 81.5 | 91.9 | 96.6 | 97.7 |
| $||\mathcal{M}[i] - \mathcal{M}[j]||_2^2$ in Eqn. 8 | 83.1 | 92.5 | 97.0 | 98.0 |
| w/o dynamically weighting | 85.0 | 93.5 | 97.3 | 98.3 |
| BMW | **86.3** | **94.2** | **97.7** | **98.4** |

It can be observed from Table 2 that the proposed inter-class constraint effectively boosts the performance by a large margin, *e.g.*, improving the mAP from 81.5% to 86.3%. It is worth noting that the improvement of $\mathcal{L}_{inter}$ is based on a strong baseline that achieves 81.5% mAP. This indicates that the separability of memory banks is important for discriminative ReID feature learning, which is neglected in previous methods. This also demonstrates that the proposed inter-class constraint is able to effectively enhance the ReID feature optimization by completing the constraints of the objective function for the memory bank rewriting.

The proposed novel formulation in Eqn. 8 for the inter-class constraint also enhances the performance as shown in Table 2. For example, it improves the mAP from 83.1% to 86.3% by 3.2% compared with the previous used $||\mathcal{M}[i] - \mathcal{M}[j]||_2^2$ version. The comparison results clearly demonstrate that, with only one update step, the proposed formulation in Eqn. 8 is more effective in enlarging the angle between two L2 normalized features in memory banks.

As shown in Table 2, the proposed dynamically weighting strategy further boosts the accuracy of the ReID model from a rather high performance, *e.g.*, it boosts the mAP accuracy from 85.0% by 1.3%. Note that, compared to "baseline [22]", "w/o $\mathcal{L}_{inter}$" only additionally uses the proposed dynamically weighting strategy and achieves a better performance, *e.g.*, 81.5% *v.s.* 79.7%. This demonstrates the generalizability of the proposed dynamically weighting strategy that consistently enhances the memory bank rewriting for both inter-class and intra-class constraints.

Table 3: Comparison with SOTA methods on Market-1501 and MSMT17. "S. D." denotes the used labeled source data. "‡" denotes multi-scale features are used. "†" denotes extra camera label is used.

| Method | Backbone | Market-1501 | | | | | MSMT-17 | | | | |
| --- | --- | --- | --- | --- | --- | --- | --- | --- | --- | --- | --- |
| | | S. D. | mAP | Rank1 | Rank5 | Rank10 | S. D. | mAP | Rank1 | Rank5 | Rank10 |
| UTAL [41] | ResNet50 | - | 46.2 | 69.2 | - | - | - | - | - | - | - |
| SpCL [17] | ResNet50 | - | 73.1 | 88.1 | 95.1 | 97.0 | - | - | - | - | - |
| GCMT [37] | ResNet50 | - | 73.9 | 89.7 | 96.5 | 97.6 | - | 23.7 | 54.3 | - | - |
| SSG‡ [43] | ResNet50 | - | 58.3 | 80.0 | 90.0 | 92.4 | Market | 13.2 | 31.6 | - | - |
| CR_GAN[19] | ResNet50 | Duke | 54.0 | 77.7 | 89.7 | 92.7 | - | - | - | - | - |
| PDA-Net [20] | ResNet50 | Duke | 47.6 | 75.2 | 86.3 | 90.2 | - | - | - | - | - |
| DIM+GLO [18] | ResNet50 | Duke | 65.1 | 88.3 | 94.7 | 96.3 | Market-1501 | 20.7 | 49.7 | 66.1 | - |
| UDA [36] | ResNet50 | Duke | 53.7 | 75.8 | 89.5 | 93.2 | - | - | - | - | - |
| MMT [38] | ResNet50-IBN | Duke | 76.5 | 90.9 | 96.4 | 97.9 | - | - | - | - | - |
| MEB-Net [39] | DeseNet121 | - | 76.0 | 89.9 | 96.0 | 97.5 | - | - | - | - | - |
| MMT [38] | ResNet50 | - | 71.2 | 87.7 | 94.9 | 96.9 | Duke | 29.7 | 58.8 | 71.1 | 76.1 |
| IIDS [65] | - | 78.0 | 91.2 | 96.2 | 97.7 | - | 35.1 | 64.4 | 76.2 | 80.5 | |
| PPLR [45] | ResNet50 | - | 81.5 | 92.8 | 97.1 | 98.1 | - | 31.4 | 61.1 | 73.4 | 77.8 |
| PPLR"†" [45] | ResNet50 | - | 84.4 | 94.3 | 97.8 | 98.6 | - | 42.2 | 73.3 | 83.5 | 86.5 |
| CaCL [2] | ResNet50 | MSMT17 | 84.7 | 93.8 | 97.7 | 98.1 | Market-1501 | 36.5 | 66.6 | 75.3 | 80.1 |
| DCC [23] | ResNet50-IBN | - | 58.8 | 94.3 | 97.6 | 98.6 | - | 36.6 | 64.9 | 74.9 | 78.5 |
| IICS [66] | ResNet50 | - | 72.1 | 88.8 | 95.3 | 96.9 | - | 18.6 | 45.7 | 57.5 | 62.8 |
| IICS [66] | ResNet50 | - | 72.9 | 89.5 | 95.2 | 97.0 | - | 29.9 | 56.4 | 68.8 | 73.4 |
| O2CAP+FuseDSI [67] | ResNet50 | - | 83.4 | 93.3 | 97.2 | 98.3 | - | 44.2 | 73.7 | 83.6 | 86.7 |
| ISE [68] | ResNet50 | - | 84.7 | 94.0 | 97.8 | 98.8 | - | 35.0 | 64.7 | 75.5 | 79.4 |
| **BMW** | ResNet50 | - | **86.3** | **94.2** | **97.7** | **98.4** | - | **44.6** | **75.5** | **86.3** | **87.1** |

To further illustrate the effectiveness of each component in this paper, we compare the training procedures of the ablation study in Fig. 5. It can be observed that different compared methods perform similarly in the start stage of the training. When models achieve a rather high performance, different compared methods show different abilities to further boost the performance. It can be observed that the method without the intra-class constraint performs the worst, which indicates that ensuring the separability of memory banks via the proposed intra-class constraint enables the model to gain additional discrimination capacity for the ReID task. It can also be observed that, compared with the previously used Euclidean distance in the inter-class constraint, the proposed new formulation in Eqn. 8 is more effective for memory bank rewriting within one step.

We also evaluate different sampling strategies including random sampling, average sampling and easy sampling for computing Eqn. 4. The training procedures w.r.t. different strategies are shown in Fig. 6. It can be observed that easy sampling that select easy sample for memory rewriting performs the worst, which is due to limited information gain is delivered in those easy samples. The random sampling strategy performs similar with the hard sampling strategy because it can also selects hard samples at times. It clear that the hard sampling strategy performs the best through the entire training procedure.

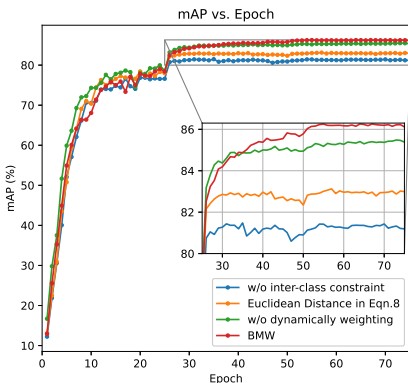

Figure 5: Comparison of training procedures.

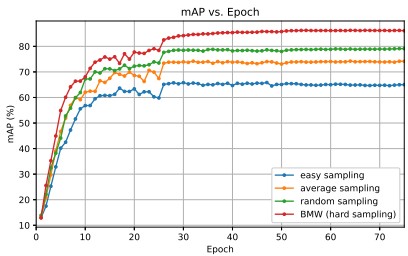

Figure 6: Comparison of sampling strategies.

## 5.4 Comparison with SOTA Methods

The comparison results one both Market-1501 and MSMT17 with SOTA methods are summarized in Table 3. It's clear that the proposed BMW outperforms existing SOTA methods by a clear margin. For example, BMW outperforms the recent SOTA method CaCL [2] by 1.6% and 2.4% in mAP on Market-1501 and MSMT17, respectively. It is worth noting that CaCL additionally uses labeled

source data for the model training. It can also be observed that BMW outperforms other domain adaptive methods that use extra labeled data, *e.g.*, DIM+GLO [18] and MMT [38], by a large margin. This indicates that BMW is more effective for discrminative feature optimization for the ReID task.

Compared with recent methods that use the momentum mechanism to rewrite memory banks [23, 22, 18, 17], the proposed BMW outperforms them by a clear margin. For instance, BMW outperforms the recent DCC by 7.5% and 8.0% in mAP on Market-1501 and MSMT17, respectively. This indicates that the inter-class constraint introduced in our BMW can effectively enhance the discrimination capacity of memory banks and therefore boosts the performance of learned features. It can also be observed that, even several methods use extra information, stronger backbone, or multi-scale features to boost the performance, the proposed BMW still outperforms them by a clear margin. This further demonstrates the effectiveness of our method.

## 6 Conclusion

This paper proposes a Bidirectionally Memory bank reWriting (BMW) method for the unsupervised person ReID task. The proposed BMW method addresses the limitations of existing memory bank rewriting algorithms that focus solely on intra-class compactness while neglecting inter-class separability. By formulating the rewriting process as a gradient descent update with dual objectives, *i.e.*, reducing intra-class diversity and enhancing inter-class separability, BMW effectively optimizes the representation and discrimination capacities of memory banks, thus provides an effective feature optimization for ReID model training. The newly designed objective formulation for the inter-class constraint further improves the separability of memory banks within a limited number of rewriting steps. Extensive experiments on standard benchmarks demonstrate the superior performance of BMW, even outperforming previous methods with stronger backbones. This simple yet effective approach establishes a new paradigm for unsupervised person ReID methods based on memory banks and has the potential to significantly advance the field of unsupervised person ReID.

**Limitations and future research**

While BMW shows promising performance, inter-class and inter-class constraints are combined with loss weights that need to be adjusted manually. As datasets could be biased, loss weights and the sampling strategy could be susceptible to replicate these biases. Therefore, for future improvement, 1) enhancing the dynamically weighting strategy to be aware of the distribution of different datasets, 2) a progressively sampling strategy w.r.t different stages of the model training, and 3) a distribution metric to guide the parameter settings, could be studied to enhance the robustness and effectiveness of the memory bank rewriting, thereby improving the performance of the trained ReID model.

**Broader Impact**

This paper innovatively formulates the memory bank rewriting as gradient decent update for objective functions, which may inspire new algorithms, theoretical, and experimental investigation. The proposed BMW method offers significant improvements on the unsupervised person ReID task and can also be used for many different vision tasks.

**Acknowledgement**

This work is supported in part by the National Key Research and Development Program of China under Grant 2024YFB4708900, in part by the National Natural Science Foundation of China under Grants U21A20486, 62473208, 92367105, and 62401294, in part by the Tianjin Science Fund for Distinguished Young Scholars under Grant 20JCJQJC00140, in part by the Postdoctoral Fellowship Program of CPSF under Grant Number GZC20240753, in part by the China Postdoctoral Science Foundation under Grant Number 2025M771529, in part by the China Postdoctoral Science Foundation - Tianjin Joint Support Program under Grant Number 2025T020TJ, and in part by the Fundamental Research Funds for the Central Universities under Grant Number 63243158 and 63253238. Dr. Jianing Li is supported by P0048887 (ITF-ITSP, ITS/028/22FP), P0051906 (RGC Early Career Scheme, 25600624), and P0054482 (Two Square Capital Limited donation).

Finally, Xiaobin Liu and Jianing Li would like to thank Xixi Wang and Yuning Zhou, respectively, for their invaluable support over the years.

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

# A  Appendix

## A.1  Proof of the advantage of Eqn. 8

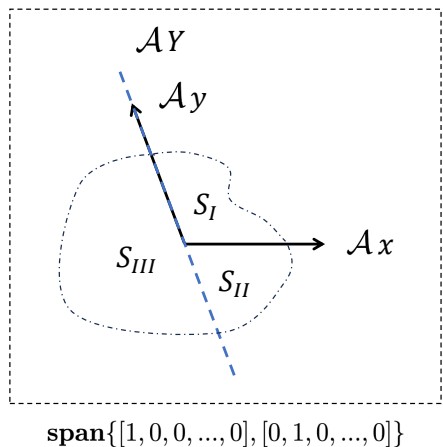

$$\mathbf{span}\{[1, 0, 0, ..., 0], [0, 1, 0, ..., 0]\}$$

Figure 7: Plot of $\mathcal{A}Y$, $\mathcal{A}x$ and $\mathcal{A}y$ in $\mathbf{span}\{[1, 0, 0, ..., 0], [0, 1, 0, ..., 0]\}$

**Theorem 1.** *Let $x, y \in \mathbb{R}^n$ be L2 normalized, i.e., $\|x\|_2 = \|y\|_2 = 1$. Consider $0 < \lambda < 1$ and the following vectors*

$$a = (1 + \lambda)x - \lambda y, \text{(update by previous methods)}$$
$$b = (1 - \lambda)x - \lambda y. \text{(update by Eqn. 8)} \qquad (12)$$

*Let $\theta_{(y,a)}$ and $\theta_{(y,b)}$ be respectively the angle between $y$ and $a$ and that between $y$ and $b$. Then, $\theta_{(y,b)} \geq \theta_{(y,a)}$.*

*Proof.* We only need to consider the case where $y \neq x$. Let X be the space expanded by the vector $x$, $\mathrm{proj}(y, X)$ be the projection of $y$ on $X$ and $y_\perp = y - \mathrm{proj}(y, X) \neq 0$. Since $x$ and $y_\perp / \|y_\perp\|$ are orthogonal and normalized, there exists an orthonormal basis expanded by these two vectors. Therefore, there exists one isometry $\mathcal{A}$ such that $\mathcal{A}x = [1, 0, 0, ..., 0] \in \mathbb{R}^n$ and $\mathcal{A}(y_\perp / \|y_\perp\|) = [0, 1, 0, ..., 0] \in \mathbb{R}^n$. Recall that isometry preserves norms and angles. In the following, we focues on $\theta_{(\mathcal{A}y, \mathcal{A}b)}$, $\theta_{(\mathcal{A}y, \mathcal{A}a)}$ and show that $\theta_{(\mathcal{A}y, \mathcal{A}b)} > \theta_{(\mathcal{A}y, \mathcal{A}a)}$.

Since $y = \mathrm{proj}(y, X) + y_\perp$ and $\max(\|y_\perp\|, \|\mathrm{proj(y, X)}\|) < 1$, there exist $-1 < \alpha < 1$ and $0 < \beta < 1$ such that $\alpha^2 + \beta^2 = 1$

$$\mathcal{A}y = [\alpha, \beta, 0, ....0]$$

By restricting on the subspace $\mathbf{span}\{[1, 0, 0, ..., 0], [0, 1, 0, ..., 0]\}$ and letting $Y = \mathbf{span}\{y\}$, we plot $\mathcal{A}Y$, $\mathcal{A}x$ and $\mathcal{A}y$ in Figure 7. We also characterize the three sets $S_I$, $S_{II}$, $S_{III}$, which have boundaries defined by $\mathcal{A}Y$, $\mathcal{A}x$ and $\mathcal{A}y$ as shown in Figure 7:

$$S_I = \{\mu \mathcal{A}x + \nu \mathcal{A}y : \mu, \nu > 0\}$$
$$S_{II} = \{\mu \mathcal{A}x - \nu \mathcal{A}y : \mu, \nu > 0\}$$
$$S_{III} = \{v \in \mathbb{R}^n : \langle v, [\beta, -\alpha, 0, ..., 0] \rangle < 0\}$$

One can verify that $\mathcal{A}a$ and $\mathcal{A}b$ are both in the set $S_{II}$. Therefore, by denoting $\theta_b = \theta_{(\mathcal{A}x, \mathcal{A}b)}$ and $\theta_a = \theta_{(\mathcal{A}x, \mathcal{A}a)}$, we have $0 < \theta_b, \theta_a < \pi$

$$\theta_{(\mathcal{A}y, \mathcal{A}b)} = \theta_{(\mathcal{A}y, \mathcal{A}x)} + \theta_b,$$
$$\theta_{(\mathcal{A}y, \mathcal{A}a)} = \theta_{(\mathcal{A}y, \mathcal{A}x)} + \theta_a.$$

In the following, we only need to show that $\theta_b > \theta_a$. We have,

$$\tan \theta_b = \frac{\lambda \beta}{1 - \lambda - \lambda \alpha}, \quad \tan \theta_a = \frac{\lambda \beta}{1 + \lambda - \lambda \alpha}.$$

Since $1 + \lambda - \lambda \alpha > 0$, we consider two cases. For the first case, i.e., if $1 - \lambda - \lambda \alpha <= 0$, then $\theta_b > \theta_a$ because $\tan \theta_b < 0$ (i.e., $\theta_b > \pi/2$) or $\tan \theta_b$ is not well defined (i.e., $\theta_b = \pi/2$) while $\tan \theta_a > 0$ (i.e., $\theta_a < \pi/2$). For the second case, i.e., if $1 - \lambda - \lambda \alpha < 0$, then $\theta_b > \theta_a$ because $\tan \theta_b > \tan \theta_a > 0$.

$\square$

## A.2 Details of Gradient in Eqn. 11

$$
\begin{aligned}
\frac{\partial \mathcal{L}_{intra}(\mathcal{M}[c], f_i)}{\partial \mathcal{M}[c]} &= \frac{1}{8} \times 4 \times \|\mathcal{M}[c] - f_i\|_2^3 \times \frac{\mathcal{M}[c] - f_i}{\|\mathcal{M}[c] - f_i\|_2} \\
&= \frac{1}{2}\|\mathcal{M}[c] - f_i\|_2^2 (\mathcal{M}[c] - f_i) \\
&= \frac{1}{2}(\|\mathcal{M}[c]\|_2^2 + \|f_i\|_2^2 - 2(\mathcal{M}[c])^\top f_i)(\mathcal{M}[c] - f_i) \\
&= \frac{1}{2}(1 + 1 - 2(\mathcal{M}[c])^\top f_i)(\mathcal{M}[c] - f_i) \\
&= (1 - (\mathcal{M}[c])^\top f_i)(\mathcal{M}[c] - f_i).
\end{aligned}
$$

$$
\begin{aligned}
\frac{\partial \mathcal{L}_{inter}(\mathcal{M}[i], \mathcal{M}[j])}{\partial \mathcal{M}[i]} &= \frac{1}{8} \times 4 \times \|\mathcal{M}[i] + \mathcal{M}[j]\|_2^3 \times \frac{\mathcal{M}[i] + \mathcal{M}[j]}{\|\mathcal{M}[i] + \mathcal{M}[j]\|_2} \\
&= \frac{1}{2}\|\mathcal{M}[i] + \mathcal{M}[j]\|_2^2 (\mathcal{M}[i] + \mathcal{M}[j]) \\
&= \frac{1}{2}(\|\mathcal{M}[i]\|_2^2 + \|\mathcal{M}[j]\|_2^2 + 2(\mathcal{M}[i])^\top \mathcal{M}[j])(\mathcal{M}[i] + \mathcal{M}[j]) \\
&= \frac{1}{2}(1 + 1 + 2(\mathcal{M}[i])^\top \mathcal{M}[j])(\mathcal{M}[i] + \mathcal{M}[j]) \\
&= (1 + (\mathcal{M}[i])^\top \mathcal{M}[j])(\mathcal{M}[i] + \mathcal{M}[j]).
\end{aligned}
$$

## A.3 Experiments with different backbones

Currently, ResNet50 is the most widely used backbone for unsupervised person ReID. Therefore, we adopt it in our paper ensuring fair comparisons with other approaches. To further validate our method's generality, we conduct additional experiments with ResNet18, ResNet34, and ViT-B/16. Key results on Market-1501 are summarized in Table 4, results on DukeMTMC-reID are provided on the repository on github.

Table 4: Performance comparison on Market-1501 with different backbones.

| Backbone | Method | mAP |
|---|---|---|
| ResNet18 | w/o $\mathcal{L}_{inter}$ | 57.1 |
|  | BMW | **67.9** |
| ResNet34 | w/o $\mathcal{L}_{inter}$ | 65.4 |
|  | BMW | **71.1** |
| ViT-B/16 | w/o $\mathcal{L}_{inter}$ | 84.6 |
|  | BMW | **87.6** |

The mAPs yielded with ResNet18 and ResNet34 are lower than those with ResNet50 due to the smaller structures of the former two networks. The performances with ViT-B/16 are slightly better than that with ResNet50 because of its stronger capacity. We do not adjust hyper-parameters with those backbones and only evaluate our core contribution, *i.e.*, the inter-class constraint.

It's clear that, with different backbones, the proposed inter-class constraint can always boost the ReID performance. These experimental results show that our BMW method could generalize well with different backbone. This is because our method is proposed to update the memory bank with L2-normalized features and it is backbone-agnostic. Thus, our BMW can be applied to any backbones without modification.

The experimental results demonstrate that the proposed inter-class constraint consistently enhances ReID performance across all tested backbones, confirming the strong generalizability of our BMW method. This adaptability stems from our backbone-agnostic design: the memory bank is updated with L2-normalized features. Therefore, BMW seamlessly integrates with any backbone architecture without requiring modifications.

## A.4 Implementation details

We use the unlabeled training set of target domain to train the model, and use the testing set of target domain to evaluate the model. DBSCAN is used for the unsupervised clustering and the eps is set as 0.6 following [22]. Input images are resized to $256\times128$. We use random flipping, random cropping, and random erasing [69] for data augmentation. The Adam optimizer is adopted for training. Learning rate is initialized as 0.00035 and

decayed by 0.1 every 25 epochs. Model is totally trained for 75 epochs. Model is trained on a server with 4 RTX 4090 GPU, and 128G memory.

In each training batch, we randomly sample 16 identities, *i.e.*, clusters by DBSCAN. For each sampled identity, 16 images are randomly selected to compose the training batch, resulting 256 images. For inter-class constraint computation ($\mathcal{L}_{inter}$), we update the 16 memory banks corresponding to those sampled 16 identities by their closest memory bank vector (hard sampling strategy) via $\mathcal{L}_{inter}$.

Compared with the baseline model, our BMW method introduces two extra steps:

1. Finding the closest memory bank vector for 16 memory banks. This costs less than 1ms for the memory banks with a size of less than 10000 (only around 1000 in our experiments on person ReID datasets) with the Intel(R) Core(TM) i7-14700 CPU.

2. Computing the update vector for the 16 memory banks by $\mathcal{L}_{inter}$. As shown in Eqn. (11), computing the update vector for each memory bank only introduces one vector dot-product, one vector addition, one vector scaling, and one scalar addition. This only needs 4096 multiplications and 2049 additions with 2048 feature dimensions in our paper and also costs less than 1ms with CPU (will be faster with GPU).

Therefore, our method only introduces negligible overhead compared with the baseline method. We also test the training time overhead: training the baseline model costs 3h25m33s, whereas ours costs 3h26m53s (just an extra 80s over roughly 3.5h).

As our method only updates the memory bank used in the training process and only the backbone is used to extract image features in the inference process, our method does not affect the inference efficiency.

The dynamic weighting could capture the overall distribution because:

1. For the intra-class constraint, 16 samples are randomly sampled for each cluster, which could be enough to provide an approximate reflection of the entire sample distribution in this cluster. For instance, there are roughly 20 samples in each cluster in the training process on Market-1501.

2. For the inter-class constraint, the hard negative sample is selected from all memory banks of the dataset, *i.e.*, it is class-wise. Thus, it can also sense the overall data distribution.

## A.5  Experiments on VeRi-776

VeRi-776 contains 37,746 images of 576 vehicles for training, 1,678 images and 51,003 images of another 200 vehicles are used as queries and galleries, respectively. The comparison results on VeRi-776 with other methods are summarized in Table 5.

Table 5: Comparison with SOTA methods on VeRi-776.

| Method | Reference | Source Data | mAP |
|---|---|---|---|
| MMT [38] | ICLR 2020 | VehicleID | 35.3 |
| SpCL[17] | NeurIPS 2020 | VehicleID | 38.9 |
| Cluster Contrast [22] | ACCV 2022 | - | 40.3 |
| Cluster Contrast (Infomap) [22] | ACCV 2022 | - | 40.8 |
| DCC [23] | arxiv 2022 | - | 41.4 |
| PPLR (part feature) [45] | CVPR 2022 | - | 41.6 |
| BMW | This paper | - | **42.1** |
| DCC (ResNet50-ibn) [23] | arxiv 2022 | - | 42.1 |
| PPLR(part feature+camera labels) [45] | CVPR 2022 | - | 43.5 |
| PPLR(part feature+camera labels+re-ranking) [45] | CVPR 2022 | - | 45.3 |

It can be observed from Tabel 5 that our BMW method also outperforms other methods on the vehicle ReID task. For example, PPLR (part features) uses extra part features and our BMW still outperforms it by 0.5% in mAP. Our BMW method is even comparable with methods that use deeper backbone [DCC (ResNet50-ibn)], extra annotations [PPLR (part features + camera labels)] or stronger re-ranking PPLR (part features + camera labels + re-ranking)].

As we use the same hyper-parameters on the VeRi-776 dataset, *i.e.*, $\lambda_{intra} = 0.9$ and $\lambda_{inter} = 0.2$, experiments on VeRi-776 further show that hyper-parameters selected on one dataset can be applied on other datasets with promising performance.

### A.6 Balance between the intra-class and inter-class constraints

The two constraints are balanced with two loss weights, *i.e.*, $\lambda_{intra}$ for the intra-class constraint $\mathcal{L}_{intra}$, and $\lambda_{inter}$ for the inter-class constraint $\mathcal{L}_{inter}$. The values of $\lambda_{intra}$ and $\lambda_{inter}$ should be different due to their distinct roles:

1. The term $\mathcal{L}_{intra}$ encourages memory banks to approach corresponding positive training samples, a larger $\lambda_{intra}$ enhances the intra-class constraint and encourages the memory banks to closely represent their positive samples while maintaining discriminative power in feature space.

2. In the meantime, since $\mathcal{L}_{inter}$ pushes memory banks away from each other, a larger $\lambda_{inter}$ may push memory banks far away from corresponding positive samples.

Therefore, $\lambda_{inter}$ should be smaller than $\lambda_{intra}$ from the above analysis.

We also conduct the hyper-parameter analysis, which shows that:

1. The configuration ($\lambda_{intra} = 0.9$, $\lambda_{inter} = 0.2$) achieves peak performance on both datasets. This indicates that hyperparameters selected on one dataset can be applied to others without sacrificing the performance.

2. The superior performance does not totally rely on careful selection of hyperparameters. We notice that BMW consistently achieves better results than others as long as $\lambda_{intra}$ sits in $[0.6, 1]$ and $\lambda_{inter}$ in $[0.1, 0.25]$.

Consequently, the hyperparameters are easily tunable for superior performance across different datasets.

By the way, the superiority range of $\lambda_{inter}$ is smaller than that of $\lambda_{intra}$. This is because $\mathcal{L}_{intra}$ decreases after several training iterations even when using the hard sampling strategy, while $\mathcal{L}_{inter}$ stays large because of the large number of identities. We can apply a threshold on $\mathcal{L}_{inter}$ to ignore memory bank pairs that are already far away from each other. This strategy will enlarge the operating range of $\lambda_{inter}$.

### A.7 Limitations

#### A.7.1 Noisy data

Our method is applied to the unsupervised person ReID task. As the pseudo label is generated by unsupervised clustering method, *i.e.*, DBSCAN, the training process is performed with noisy in the label. Along with the model training, the discrimination ability of the model is progressively enhanced. The pseudo label generated by DBSCAN based on extracted features will be also progressively more precise, which in turn helps the model training. However, the noisy label will mislead both the model training and the memory bank optimization. Therefore, for future improvement, a robust memory bank optimization method should be studied that better captures relationships between memory entries and training samples to improve effectiveness.

#### A.7.2 Cross-domain scenarios

Our experiments demonstrate both strong performance and consistent hyperparameter transferability across datasets. While current unsupervised person ReID methods typically train and test on the same dataset, cross-domain evaluation reveals significant performance degradation due to distribution shifts.

Recent work in lifelong person ReID has begun addressing this challenge through memory bank-based knowledge preservation. Current approaches primarily optimize memory banks using only positive sample relationships. We observe that incorporating inter-class constraints, as in our BMW framework, could significantly enhance knowledge representation. Building on this insight, future research should investigate BMW-based memory bank optimization to improve performance in lifelong ReID scenarios. This direction could potentially bridge the gap between single-domain and cross-domain ReID performance.

#### A.7.3 Optimization effectiveness

For datasets with a large number of identities, the inter-class loss may remain elevated during training. Therefore, we need to adjust inter-class constraints. For future research, a dynamically weighting strategy and a progressively hard sampling strategy could be studied to enhance the robustness and effectiveness of our method.

### A.8 Compared with the conventional classifier

It's an important topic about memory bank is better than FC layer for unsupervised person ReID. We provide the reasons as follows.

Memory bank is more effective in enhancing feature representation:

1. ReID is different from classification: In the classification task, the training and test sets have the same set of categories. However, in ReID task, identities in the training set do not exist in the test set (because we cannot know identities appear in the future). So, a classifier trained on the training set can not predict correct ID labels on the test set. As ReID is performed based on image feature similarities, feature distance relationship is more important than classification prediction.

2. Classifier is not effective to optimize features: The standard classifier uses an FC layer and a SoftMax layer to predict probabilities. The cross-entropy loss is commonly used to supervise the predicted probabilities. Classifier focuses on predicting correct labels, instead of optimizing feature distance relationships, as widely discussed in previous works.

3. Memory bank is more effective to optimize features: The memory bank is optimized separately with the features. The target of memory bank update is to enhance the feature distance optimization and the performance of ReID, rather than accurately predicting image labels. *E.g.*, although $\mathcal{L}_\theta$ increases from 0.439 to 3.033 in the epoch 1 on Market, mAP on test set increases from 12.9% to 25.6%.

Memory bank is more robust against label noise induced by unsupervised clustering:

1. A single noisy sample will perturb all parameters in FC layer: As an FC is followed by the SoftMax layer, each noisy sample will perturb all FC parameters via the gradient of the cross entropy loss, making FC parameters sensitive to label noise.

2. Only a few memory banks are perturbed by a single noisy sample: For a noisy sample in a training batch, only its corresponding memory bank is certainly to be perturbed. Other memory banks are perturbed only if the noisy label is selected as the hard sample, which has a very low probability of occurrence given the large number of identities in datasets. Thus, most memory banks will not be perturbed by a single noisy sample.

Therefore, memory banks are better for unsupervised ReID with stronger generalization capacity on test set.

On the supervised ReID task, memory bank can also enhance feature representations more effectively compared to conventional classifiers. Below, we provide the detailed explanation and validation.

BMW is more effective at feature optimization:

1. Different from classification tasks, ReID requires learning discriminative feature representations where feature distance relationships (*e.g.*, intra-class compactness and inter-class separation) are optimized, rather than classification accuracy.

2. As conventional classifiers focus on accurate label prediction and do not directly optimize feature relationships, their abilities to optimize feature representations are limited.

3. Compared to conventional classifiers, the proposed BMW directly optimizes feature distance relationships with $\mathcal{L}_\theta$, and memory banks are optimized with $\mathcal{L}_\mathcal{M}$ to facilitate the feature representation learning.

From the above analysis, the advantage of our BMW in feature representation learning over conventional classifiers is independent of data annotations. Thus, BMW is also more effective at optimizing feature representations on the supervised ReID task compared to conventional classifiers.

We further trained our BMW method under the supervised setting. The performance comparison between our BMW and conventional classifiers is summarized in Table 6.

Table 6: Comparison among classifier-based and memory-bank-based methods.

| Method | Type | Reference | mAP on Market | mAP on MSMT17 |
|---|---|---|---|---|
| Only classifier | Classifier | IJCAI'21 | 81.1 | – |
| Classifier+circle loss | Classifier | CVPR'20 | 87.4 | 52.1 |
| ICE | Memory bank | ICCV'21 | 86.6 | 50.4 |
| ISE | Memory bank | CVPR'22 | 87.8 | 51.0 |
| DCMIP | Memory bank | ICCV'23 | 89.2 | 62.8 |
| DCC | Memory bank | arXiv'22 | 89.9 | 65.5 |
| **Unsupervised BMW** | Memory bank | This paper | 86.3 | 44.6 |
| **Supervised BMW** | Memory bank | This paper | **90.5** | **66.7** |

It can be observed from Table 6 that:

1. Compared to the conventional classifier, our BMW still enhances feature representations more effectively under the supervised setting.

2. Our unsupervised version of BMW is even comparable to the supervised circle loss method, which is a metric learning method. This shows the stronger feature optimization capacity of our BMW method.

3. Our BMW also outperforms other methods using memory bank by a clear margin. This further shows the superiority of our BMW method.

