# OpenReview forum: "BMW: Bidirectionally Memory bank reWriting for Unsupervised Person Re-Identification"
_NeurIPS.cc/2025/Conference — NeurIPS 2025 poster_

### Official Review · Reviewer_dA3y · 2025-06-07

**Clarity:** 4
**Significance:** 3
**Originality:** 3
**Rating:** 4
**Confidence:** 4

**Summary:**

A novel method, Bidirectionally Memory bank reWriting (BMW), is proposed for unsupervised person reID. The overall structure is based on the Memory Bank approach for unsupervised person reID through contrastive learning. The key problem to be tackled is that existing approaches with the memory bank contrastive learning mainly focus on intra-class compactness. This paper introduces the inter-class constraint into the loss for the model training. In such a way, it would improve the feature discriminability. Instead of the common momentum mechanism used in the memory bank update, this paper redefines the memory bank update through a gradient descent update. Experiments show that BMW achieves state-of-the-art performance on benchmarks like Market-1501 and MSMT17.

**Questions:**

1. Training oscillations could be an issue. In a single batch, due to training samples coming from the different classes, the memory bank update in each round may be affected by multiple negative classes. At the same time, these classes may have an impact on each other.  This could cause the memory update to be unstable. If the data class imbalance and fake negative (caused by label noise) exist, such an issue could be more serious.

2. Current dynamic weighting is on the batch level. It cannot sense the overall distribution of the whole dataset. If a certain class is always "easy" but not "hard" sample. The corresponding class information inside the memory bank may not be optimised properly.

3. In Equation 4, Lambda_intra and Lambda_inter settings are sensitive to the performance. For example, larger lambda_inter could cause training to be unstable. A more robust/dynamic way for determining these two parameters will be helpful.

4. The equation 10 is a function of the power of 4. It seems that equation 11 (differentiation of equation 10) is wrong. I cannot see the right derivation over here.

5. There is no explanation on Fig 2.

6. There are some typos on line 226.

7. In line 339, it said BMW outperforms CaCL by 2.4% on MSMT17 in terms of mAP. This is different from the information on Table 3.

**Ethical Concerns:**

["NO or VERY MINOR ethics concerns only"]

**Final Justification:**

The authors have addressed the key concerns raised in my reviews. I am happy to lift the scores from "3: Borderline reject" to "4: Borderline accept".

**Limitations:**

See the comments above in "Weaknesses" and "Questions"

**Paper Formatting Concerns:**

No concerns.

**Quality:**

3

**Strengths And Weaknesses:**

Strengths:

1) This paper introduces inter-class constraints into the contrastive learning for memory bank update through gradient descent.
2) The power of 4 of Euclidean distances (Equation 10) is introduced to measure the positive pair and the negative pair in the contrastive learning for updating the loss in the intra-class constraint loss and the inter-class constraint loss. This update helps dynamically update the weighting to reflect the importance of each training sample on its contribution to memory bank update. It will make the training focus on the training samples causing larger differences (i.e. hard samples).
3) Comprehensive ablation studies are introduced with reasonable visualisations and theoretical proof provided.

Weaknesses:

1) The dynamic weighting introduced in this paper is to reflect the different importance of each training sample on updating the memory bank. However, in terms of the loss function design (Equation 4), the weightings are manually tuned and not automatically adapted based on data distribution or training dynamics. This could be an issue if the dataset has the problem of class imbalance.
2) The proposed BMW should have a contribution to any model based on Memory Bank through contrastive learning. It will be helpful to carry out other experiments beyond person reID application.
3) There should be more new methods in the past 2 years (ie 2024 & 2025) for comparison. Or some more popular method will be appreciated such as Cluster Contrast.

---

> ### Author Rebuttal · Authors · 2025-07-31
>
> **Response to the Reviewer dA3y**
>
> ----
>
> We sincerely appreciate your recognition of the novelty of this paper. And we thank you very much for detailed and constructive comments.
> The point-by-point detailed response and the revision plans to each of your concern are given as follows:
>
> ---
> **Q1 & Q3:** Training oscillations and settings of loss weights.
>
> **A1:** Thanks for the insightful question! We would like to respond to your Q1 on training oscillations and Q3 on the loss weight settings together.
>
> To chase a stable training, we use two loss weights $\lambda _ {intra}$ and $\lambda _ {inter}$ for intra-class and inter-class constraints to control the scale of memory bank update.
>
> Theoretical analysis indicates that $\lambda _ {intra}$ could be a large value and $\lambda _ {inter}$ should be small. The reasons are:
> - Because $\mathcal{L} _ {intra}$ encourages memory banks to approach corresponding positive training samples, a larger $\lambda _ {intra}$ will emphasize the intra-class constraint and make memory banks similar to those positive samples, but still discriminative in the feature space.
> - While, $\mathcal{L} _ {inter}$ pushes memory banks away from each other, and a larger $\lambda _ {inter}$ may push memory banks far away from corresponding positive samples.
>
> Our method can also be regarded as an enhancement of the intra-class constraint. Therefore, the selection of loss weights can start from $\lambda _ {intra}=1$ and $\lambda _ {inter}=0.1$, where our BMW achieves 84.9% mAP on Market-1501, already outperforming most other methods. So, it's not very difficult to select the loss weights in our method. And like any other gradient-based optimization method, an excessively large loss weight can lead to model collapse.
>
> As shown in the experiments on Market-1501 (in our paper) and DukeMTMC-reID (in our response to Reviewer a7FY), the superior performance **does not totally rely on careful selection of loss weights**. We notice that BMW consistently achieves promising performance as long as $\lambda_{intra}$ sits in $[0.6,1]$ and $\lambda_{inter}$ in $[0.1,0.25]$.
>
> About class imbalance: In each training batch, we randomly sample 16 identities from the dataset, *i.e.*, clusters by DBSCAN. For each sampled identity (cluster), 16 images are randomly sampled from its cluster to compose the training batch. This sampling strategy ensures that every identity (cluster), regardless of its original cluster size, contributes equally to each mini-batch, thereby alleviating the class-imbalance issue.
>
> About noisy label: For the unsupervised person ReID task, the pseudo label generated by DBSCAN contains certain noise. Along with the model training, the discrimination ability of the model is progressively enhanced. The pseudo label generated by DBSCAN based on extracted features will be also progressively more precise, which in turn helps the model training. The noisy label does mislead both the model training and the memory bank optimization, however, the experiments show that it's fine for these datasets. For future improvement on larger datasets, a robust memory bank optimization method with considering more relationships among memory banks and training samples could be studied to enhance the robustness against label noise.
>
> Additionally, as we have discussed in the section "Limitations and future research", the following methods for selecting loss weights could be studied in the future to enhance the robustness and effectiveness of the memory bank rewriting:
>  1. Enhancing the dynamically weighting strategy to be aware of the distribution of different datasets.
>  2. A progressively sampling strategy w.r.t different stages of the model training.
>  3. A distribution metric to guide the loss weight settings.
>
> ---
>
> **Q2:** Dynamic weighting and optimization for "easy" memory banks.
>
> **A2:** Thanks for the constructive comments. We clarify that, the dynamic weighting is not totally on the batch level.
>
> We would like to first describe the detailed steps of intra-class and inter-class constraints as follows:
> 1. An unsupervised clustering method, *e.g.*, DBSCAN in our paper, is applied to cluster all the samples into different clusters based on extracted features before model training.
> 2. A memory bank is initialized for each cluster with the average vector of all L2-normalized features in this cluser and a L2 normalization.
> 3. In each training batch, we randomly sample 16 identities from the dataset, *i.e.*, clusters by DBSCAN. For each sampled identity (cluster), 16 images are randomly sampled from its cluster to compose the training batch.
> 4. 16 memory banks corresponding to those sampled 16 identities (clusters) are updated by intra-class and inter-class constraints as:
>     - For the intra-class constraint ($\mathcal{L} _ {intra}$), each memory bank is updated with its hard positive sample (the sample exhibiting  the smallest similarity to the memory bank, selected from the 16 samples in the training batch) via Eqn.(11).
>     - For the inter-class constraint ($\mathcal{L} _ {inter}$), each memory bank is updated with its hard negative memory bank vector (the memory bank vector exhibiting the largest similarity to this memory bank, selected from all the memory banks except this one) via Eqn.(11).
>
> The dynamic weighting could capture the overall distribution because:
> - For the intra-class constraint, 16 samples are randomly sampled for each cluster, which could be enough to provide an approximate reflection of the entire sample distribution in this cluster. For instance, there are roughly 20 samples in each cluster in the training process on Market-1501.
> - For the inter-class constraint, the hard negative sample is selected from all memory banks of the dataset, *i.e.*, it is class-wise. Thus, it can also sense the overall data distribution.
>
> We would also like to clarify that, the "hard" or "easy" label for sampling in the inter-class constraint is used to describe the relationship between two memory banks, instead of a single memory bank vector.
>
> Above explanations will be added in the final version for clarity.
>
> ---
>
> **Q4:** The derivation about Eqn.(10).
>
> **A4:** Thanks for pointing out that. We provide details of the derivation for Eqn.(10) as follows:
>
>   - $\frac{\partial \mathcal{L}_{intra}(\mathcal{M}[c],f_i)}{\partial\mathcal{M}[c]}$
> $=\frac{1}{8}\times4\times\|\mathcal{M}[c]-f_i\|_2^3\times\frac{\mathcal{M}[c]-f_i}{\|\mathcal{M}[c]-f_i\|_2}$
> $=\frac{1}{2}\|\mathcal{M}[c]-f_i\|_2^2(\mathcal{M}[c]-f_i)$
> $=\frac{1}{2}(\|\mathcal{M}[c]\|_2^2+\|f_i\|_2^2-2(\mathcal{M}[c])^\top f_i)(\mathcal{M}[c]-f_i)$
> $=\frac{1}{2}(1+1-2(\mathcal{M}[c])^\top f_i)(\mathcal{M}[c]-f_i)$
> $=(1-(\mathcal{M}[c])^\top f_i)(\mathcal{M}[c]-f_i)$,
>
>
>   - $\frac{\partial \mathcal{L}_{inter}(\mathcal{M}[i],\mathcal{M}[j])}{\partial\mathcal{M}[i]}$
> $=\frac{1}{8}\times4\times\|\mathcal{M}[i]+\mathcal{M}[j]\|_2^3\times\frac{\mathcal{M}[i]+\mathcal{M}[j]}{\|\mathcal{M}[i]+\mathcal{M}[j]\|_2}$
> $=\frac{1}{2}\|\mathcal{M}[i]+\mathcal{M}[j]\|_2^2(\mathcal{M}[i]+\mathcal{M}[j])$
> $=\frac{1}{2}(\|\mathcal{M}[i]\|_2^2+\|\mathcal{M}[j]\|_2^2+2(\mathcal{M}[i])^\top \mathcal{M}[j])(\mathcal{M}[i]+\mathcal{M}[j])$
> $=\frac{1}{2}(1+1+2(\mathcal{M}[i])^\top \mathcal{M}[j])(\mathcal{M}[i]+\mathcal{M}[j])$
> $=(1+(\mathcal{M}[i])^\top \mathcal{M}[j])(\mathcal{M}[i]+\mathcal{M}[j])$.
>
> This will be added in our revised version for clarity.
>
> ---
>
> **Q5:** Explanation on Fig. 2.
>
> **A5:** Thanks! We will provide a detailed caption for Fig. 2 as follows:
>
>     "Figure 2: Illustration of the advantage in inter-class constraint of the proposed objective function in Eqn.(8) w.r.t. different angles between $\mathcal{M}[i]$ and $\mathcal{M}[j]$. Yellow arrows and blue arrows denote $\mathcal{M}[i]$ and $\mathcal{M}[j]$, respectively. Green dashed arrows and red dashed arrows denote the updated $\mathcal{M}[i]$ by the proposed objective function in Eqn.(8) and the previous method, respectively. It can be observed that the objective function in Eqn.(8) is more effective in pushing $\mathcal{M}[i]$ away from $\mathcal{M}[j]$."
>
> This will be added in the revised version for clarity.
>
> ---
>
> **Q6:** Typos in Line 226
>
> **A6:** Thanks for pointing out that! We have carefully proofread the manuscript and fixed it as:
>
>  - "In this section, we first analyze the effect of hyper-parameters $\lambda _ {intra}$, $\lambda _ {inter}$ and $\tau$ on Market-1501"
>
> Typos will be corrected in the final version.
>
> ---
>
> **Q7:** Mistake about the performance comparison.
>
> **A7:** Thanks for pointing out that. 2.4% is the mAP improvement by our BMW compared with PPLR $^ {``\dag"}$ on MSMT-17. We mistakenly took it as the one compared with CaCL. We have corrected it as:
>
> - "For example, BMW outperforms the recent SOTA method CaCL [2] by 1.6% and 8.1% in mAP on Market-1501 and MSMT-17, respectively"
>
> This will be corrected in the revised version.
>
> ---
>
> **Other issues in comments**
>
> - Other tasks beyond person ReID:
> Thanks for the constructive suggestion! We plan to evaluate our method on other tasks like vehicle ReID (waiting for its authors' response regarding the data access) and self-supervised learning in the future study. By the way, we have supplemented the experimental results on DukeMTMC-reID in our response to Reviewer a7FY.
>
> - More related works:
> Sure! We will add more recent and popular works in the revised version, *e.g.*, FUReID \[PR 2024\], IIDS \[T-PAMI 2024\], FuseDSI \[IF 2024\], RTMen \[T-IP 2023\], ISE \[CVPR 2022\], P $^2$ LR \[AAAI 2022\], and IICS \[CVPR 2021\]. By the way, the Cluster Contrast method is already cited as \[15\] and  serves as the baseline model in our paper.
>
> ---
>
> We believe that these responses and revisions will address your concerns and those updated content will significantly strengthen our paper.
>
> Thank you again for your recognition of the novelty of our paper and for your invaluable feedback.

---

> > ### Comment · Reviewer_dA3y · 2025-08-01
> >
> > Thank you for the detailed point-by-point reply to the comments raised. I will carefully take them into consideration for the final rating.

---

> ### Author Response · Authors · 2025-08-01
>
> Thanks for your time and effort. We're here to address any remaining concerns and welcome further discussion. Your insights will greatly strengthen our paper!

---

> ### Author Response · Authors · 2025-08-05
> **Validation on the  Vehicle ReID task: BMW Achieves SOTA Performance on the VeRi-776 dataset**
>
> **BMW achieves *SOTA* Performance on the Vehicle ReID Task on VeRi-776**
>
> ----
>
> Following your suggestion that "*It will be helpful to carry out other experiments beyond person reID application.*", we further evaluate our method on the vehicle ReID application and our method also outperforms other methods.
>
> We received the response from authors of the VeRi-776 dataset and got the data about 12 hours ago. We performed a quick validation of our method on VeRi-776 (which costs about 10.5 hours as VeRi-776 has more images) and our BMW method also achieves SOTA performance on VeRi-776. Comparisons with other methods are given as follows:
>
> |Method|Reference|Source Data|mAP|
> |:-|:-:|:-:|:-:|
> MMT |ICLR 2020 | VehicleID | 35.3|
> |SpCL |NeurIPS 2020 | VehicleID| 38.9|
> |Cluster Contrast  |ACC 2022|-|40.3|
> |Cluster Contrast \[*Infomap*\] |ACC 2022|-|40.8|
> |DCC(ResNet50)| arxiv 2022 |-| 41.4 |
> |PPLR(*part features*)| CVPR 2022|-|41.6|
> |MLA |MTA 2024|-|38.4|
> |BMW|This paper|-|**42.1**|
> ||
> |DCC(*ResNet50-ibn*) |arxiv 2022 |-| 42.1 |
> |PPLR(*part features* + *camera labels*)| CVPR 2022|-|43.5|
> |PPLR(*part features* + *camera labels* + *re-ranking*)| CVPR 2024|-|45.3|
> ||
>
>
> It can be observed from the experiments on VeRi-776 that:
> - Our BMW method also outperforms other methods on the vehicle ReID task. For example, although PPLR(*part features*) uses extra part features and our BMW still outperforms it by 0.5% in mAP.
> - Our BMW method is even comparable with methods that use deeper backbone \[DCC(*ResNet50-ibn*)\], extra annotations \[PPLR (*part features* + *camera labels*)\] or stronger re-ranking \[PPLR(*part features* + *camera labels* + *re-ranking*)\].
> - As we use the same hyper-parameters on the VeRi-776 dataset, *i.e.*, $\lambda _ {intra}=0.9$ and $\lambda _ {inter}=0.2$, experiments on VeRi-776 further show that hyper-parameters selected on one dataset can be applied to other datasets with promising performance.
>
> Thus, we can conclude that: (1) our BMW method is effective in model training with the memory bank and (2) hyper-parameter tuning is consistent for different datasets.
>
> ---
>
> We believe that these experiments on VeRi-776 will further address your concerns about the training oscillations and hyper-parameter selecting. The validation on the vehicle ReID task will also significantly strengthen our paper.
>
> Thank you again for your constructive feedback!

---

> > ### Author Response · Authors · 2025-08-09
> >
> > We sincerely appreciate your time and constructive feedback. Your insights are invaluable in improving the quality of our paper.

---

### Official Review · Reviewer_p7UF · 2025-07-01

**Clarity:** 3
**Significance:** 3
**Originality:** 3
**Rating:** 5
**Confidence:** 5

**Summary:**

The paper introduces a novel memory bank rewriting mechanism (BMW) aimed at enhancing the performance of unsupervised person re-identification (ReID). BMW leverages a bidirectional memory bank rewriting approach that integrates intra-class and inter-class constraints to improve the representation and discrimination capabilities of the memory bank. Specifically, BMW treats the memory bank rewriting process as a gradient descent update, optimizing both intra-class compactness and inter-class separability simultaneously. Experimental results demonstrate that BMW achieves excellent performance on standard benchmarks such as Market-1501 and MSMT17, even outperforming methods with stronger backbone networks, thus validating its effectiveness and potential in the field of unsupervised ReID.

**Questions:**

1. In Sec. 4, how do the authors ensure that the intra-class and inter-class constraints in BMW are balanced in practice? Are there any challenges in tuning hyperparameters?

2. The experimental section does not address BMW's computational complexity and model efficiency, which are critical for real-world applications. Can the authors provide a comparison of resource consumption with baseline methods?

3. Sec. 6 mentions BMW's limitations but lacks in-depth discussion. Can the authors elaborate on BMW's performance under noisy data conditions and in cross-domain transfer scenarios?

**Ethical Concerns:**

["NO or VERY MINOR ethics concerns only"]

**Final Justification:**

The authors have thoroughly addressed all my concerns, particularly regarding the balance between intra-class and inter-class constraints, computational efficiency, and robustness to noisy labels. The additional experiments showcasing the superiority of their BMW method in both unsupervised and supervised settings further strengthen the paper. Moreover, the method’s ability to optimize feature representations and generalize across diverse datasets adds substantial value. Based on these improvements and clarifications, I recommend accepting the paper.

**Limitations:**

Yes, the paper briefly mentions in Sec. 6 BMW's sensitivity to noisy data and challenges in cross-domain transfer, but the discussion remains superficial.

**Quality:**

3

**Strengths And Weaknesses:**

## 1. Paper Quality (Quality)

The paper proposes a novel memory bank rewriting mechanism (BMW) to enhance the performance of unsupervised person re-identification (ReID), demonstrating a certain level of innovation. Experimental results show that BMW performs exceptionally well on standard benchmarks (Market-1501 and MSMT17), surpassing methods with stronger backbone networks, which validates the approach's effectiveness. Through ablation studies, the paper thoroughly analyzes the roles of each component in BMW, providing robust supporting evidence. However, the theoretical derivation section is lacking, particularly in justifying the advantages of BMW, where the reasonableness of assumptions and the rigor of the derivation require further refinement. While the experimental section highlights performance gains, it omits discussion on computational complexity and model efficiency—key factors for practical applications. Additionally, the specific implementation details of BMW are not clearly described; for instance, the derivation and application of Eqn. 8 lack sufficient explanation, hindering reader comprehension.

## 2. Paper Clarity (Clarity)

The paper is well-organized with a coherent logical flow, making it easy to follow from the introduction to the experimental results. The use of figures and formulas is appropriate, effectively illustrating BMW's working principles and experimental outcomes. However, some terms and symbols lack clear definitions; for example, in Sec. 4, the definition and updating method of M[c] need more detailed elaboration. The related work section does not adequately distinguish BMW from existing methods, which diminishes the visibility of its innovation. Furthermore, the use of images in the paper is limited; additional figures could better highlight the innovation or clarify complex formulas.

## 3. Paper Significance (Significance)

BMW enhances the memory bank's representation and discrimination capabilities through a bidirectional rewriting mechanism, offering a fresh perspective for unsupervised ReID research. In scenarios where data annotation is expensive, BMW provides an effective solution, contributing to advancements in unsupervised learning. However, the paper falls short in validating BMW's generalization ability, such as its effectiveness in transfer learning across different datasets or tasks, with insufficient evidence provided. The discussion of potential challenges in practical applications, such as scalability on large-scale datasets, is also not comprehensive enough.

## 4. Paper Originality (Originality)

BMW is the first to unify intra-class and inter-class constraints in memory bank optimization, presenting an innovative mechanism. The novel design of the inter-class constraint objective function effectively boosts the memory bank's discrimination capability, showcasing its uniqueness. However, the comparison with related work is insufficiently detailed, failing to emphasize BMW's distinct contributions, which reduces the prominence of its originality. Additionally, some experimental designs and result presentations resemble existing literature, slightly undermining the innovation.

---

> ### Author Rebuttal · Authors · 2025-07-31
>
> **Response to the Reviewer p7UF**
>
> -----
> Thank you very much for your time and insightful comments.
> We sincerely appreciate your recognition of the novelty of this paper.
> The point-by-point detailed response and the revision plans to each concern are given as follows:
>
> ---
> **Q1:** Balance between the intra-class and inter-class constraints.
>
> **A1:** Thanks for the insightful question. The two constraints are balanced with two loss weights, *i.e.*, $\lambda _ {intra}$ for the intra-class constraint $\mathcal{L} _ {intra}$, and $\lambda _ {inter}$ for the inter-class constraint $\mathcal{L} _ {inter}$. The values of $\lambda _ {intra}$ and $\lambda _ {inter}$ should be different due to their distinct roles:
> - The term $\mathcal{L} _ {intra}$ encourages memory banks to approach corresponding positive training samples, a larger $\lambda _ {intra}$ enhances the intra-class constraint and encourages the memory banks to closely represent their positive samples while maintaining discriminative power in feature space.
> - In the meantime, since $\mathcal{L} _ {inter}$ pushes memory banks away from each other, a larger $\lambda _ {inter}$ may push memory banks far away from corresponding positive samples.
>
> Therefore, $\lambda_{inter}$ should be smaller than $\lambda_{intra}$ from the above analysis.
>
> We also conduct the hyper-parameter analysis on both Market-1501 (as shown in our manuscript) and DukeMTMC-reID (as shown in our response to the Reviewer a7FY).
> The analysis results on Market-1501 and DukeMTMC-reID show that:
>  1. The configuration ($\lambda _ {intra}=0.9$, $\lambda _ {inter}=0.2$) achieves peak performance on both datasets. This indicates that hyperparameters selected on one dataset can be applied to others without sacrificing the performance.
>  2. The superior performance does not totally rely on careful selection of hyperparameters. We notice that BMW consistently achieves better results than others as long as $\lambda _ {intra}$ sits in $[0.6,1]$ and $\lambda _ {inter}$ in $[0.1,0.25]$.
>
> Consequently, the hyperparameters are easily tunable for superior performance across different datasets.
>
> By the way, the superiority range of $\lambda_{inter}$ is smaller than that of $\lambda _ {intra}$. This is because $\mathcal{L} _ {intra}$ decreases after several training iterations even when using the hard sampling strategy, while $\mathcal{L} _ {inter}$ stays large because of the large number of identities. We can apply a threshold on $\mathcal{L} _ {inter}$ to ignore memory bank pairs that are already far away from each other. This strategy will enlarge the operating range of $\lambda_{inter}$.
>
> These will be added in our revised version for a clear explanation. Meanwhile, a discussion on the loss weights and sampling strategy has been provided in the section of "Limitations and future research" in our manuscript.
>
> ---
>
> **Q2:** Computational complexity and model efficiency.
>
> **A2:** Thanks for the comments. We provide the following clarification for training and inference:
>
> - Training: In each training batch, we randomly sample 16 identities, *i.e.*, clusters by DBSCAN. For each sampled identity, 16 images are randomly selected to compose the training batch. When computing the inter-class constraint ($\mathcal{L} _ {inter}$) for the training batch, 16 memory banks corresponding to those sampled 16 identities are updated by their closest memory bank vector (hard sampling strategy) via $\mathcal{L} _ {inter}$. Compared with the baseline model, our BMW method introduces two extra steps:
>   1. Finding the closest memory bank vector for 16 memory banks. This costs less than 1ms for the memory banks with a size of less than 10000 (only around 1000 in our experiments on person ReID datasets) with the Intel(R) Core(TM) i7-14700 CPU.
>   2. Computing the update vector for the 16 memory banks by $\mathcal{L} _ {inter}$. As shown in Eqn.(11), computing the update vector for each memory bank only introduces one vector dot-product, one vector addition, one vector scaling, and one scalar addition. This only needs 4096 multiplications and 2049 additions with 2048 feature dimensions in our paper and also costs less than 1ms with CPU (will be faster with GPU).
>
>   Therefore, our method only introduces negligible overhead compared with the baseline method. We also test the training time overhead: training the baseline model costs 3h25m33s, whereas ours costs 3h26m53s (just an extra 80s over roughly 3.5h).
>
> - Inference: As our method only updates the memory bank used in the training process and only the backbone is used to extract image features in the inference process, our method does not affect the inference efficiency.
>
> These will added in the final version for the clarification of the model efficiency.
>
> ---
>
> **Q3:** More in-depth discussions about the limitation.
>
> **A3:** Thanks for your suggestion. We will provide more discussions about the limitations as follows:
>
> -  Noisy data:
>    Our method is applied to the unsupervised person ReID task. As the pseudo label is generated by unsupervised clustering method, *i.e.*, DBSCAN, the training process is performed with noisy in the label. Along with the model training, the discrimination ability of the model is progressively enhanced. The pseudo label generated by DBSCAN based on extracted features will be also progressively more precise, which in turn helps the model training. However, the noisy label will mislead both the model training and the memory bank optimization. Therefore, for future improvement, a robust memory bank optimization method should be studied that better captures relationships between memory entries and training samples to improve effectiveness.
>
> - Cross-domain scenarios:
>    Our experiments on Market-1501, MSMT-17 (reported in the manuscript), and DukeMTMC-reID (detailed in Response to Reviewer a7FY) demonstrate both strong performance and consistent hyperparameter transferability across datasets. While current unsupervised person ReID methods typically train and test on the same dataset, cross-domain evaluation reveals significant performance degradation due to distribution shifts.
>
>    Recent work in lifelong person ReID has begun addressing this challenge through memory bank-based knowledge preservation. Current approaches primarily optimize memory banks using only positive sample relationships. We observe that incorporating inter-class constraints - as in our BMW framework - could significantly enhance knowledge representation. Building on this insight, future research should investigate BMW-based memory bank optimization to improve performance in lifelong ReID scenarios. This direction could potentially bridge the gap between single-domain and cross-domain ReID performance.
>
>    These discussions will be added in the revised version.
>
>
> ---
>
> **Other issues in comments**
>
> - Details for the derivation:
>   We provide details of the derivation for Eqn.(10) as follows:
>
>   - $\frac{\partial \mathcal{L}_{intra}(\mathcal{M}[c],f_i)}{\partial\mathcal{M}[c]}$
>   $=\frac{1}{8}\times4\times\|\mathcal{M}[c]-f_i\|_2^3\times\frac{\mathcal{M}[c]-f_i}{\|\mathcal{M}[c]-f_i\|_2}$
>   $=\frac{1}{2}\|\mathcal{M}[c]-f_i\|_2^2(\mathcal{M}[c]-f_i)$
>   $=\frac{1}{2}(\|\mathcal{M}[c]\|_2^2+\|f_i\|_2^2-2(\mathcal{M}[c])^\top f_i)(\mathcal{M}[c]-f_i)$
>   $=\frac{1}{2}(1+1-2(\mathcal{M}[c])^\top f_i)(\mathcal{M}[c]-f_i)$
>   $=(1-(\mathcal{M}[c])^\top f_i)(\mathcal{M}[c]-f_i)$,
>
>
>   - $\frac{\partial \mathcal{L}_{inter}(\mathcal{M}[i],\mathcal{M}[j])}{\partial\mathcal{M}[i]}$
>   $=\frac{1}{8}\times4\times\|\mathcal{M}[i]+\mathcal{M}[j]\|_2^3\times\frac{\mathcal{M}[i]+\mathcal{M}[j]}{\|\mathcal{M}[i]+\mathcal{M}[j]\|_2}$
>   $=\frac{1}{2}\|\mathcal{M}[i]+\mathcal{M}[j]\|_2^2(\mathcal{M}[i]+\mathcal{M}[j])$
>   $=\frac{1}{2}(\|\mathcal{M}[i]\|_2^2+\|\mathcal{M}[j]\|_2^2+2(\mathcal{M}[i])^\top \mathcal{M}[j])(\mathcal{M}[i]+\mathcal{M}[j])$
>   $=\frac{1}{2}(1+1+2(\mathcal{M}[i])^\top \mathcal{M}[j])(\mathcal{M}[i]+\mathcal{M}[j])$
>   $=(1+(\mathcal{M}[i])^\top \mathcal{M}[j])(\mathcal{M}[i]+\mathcal{M}[j])$.
>
>   This will be added in our revised version for clarity.
>
> - Comparison with related works:
>   We will expand the discussion of related literature in the revised version, including recent advances such as, FUReID \[PR 2024\], IIDS \[T-PAMI 2024\], FuseDSI \[IF 2024\], RTMen \[T-IP 2023\], ISE \[CVPR 2022\], P $^2$ LR \[AAAI 2022\], and IICS \[CVPR 2021\]. Additionally, we provide supplementary comparison results on the DukeMTMC-reID dataset in our response to Reviewer a7FY.
>
> - Scalability on large-scale datasets:
> About the overhead on large-scale datasets: As discussed in our response "A2", our method only introduces negligible overhead compared with the baseline model.
> About the optimization effectiveness, for datasets with a large number of identities, the inter-class loss may remain elevated during training. Therefore, we need to adjust inter-class constraints. Therefore, for future research, a dynamically weighting strategy and a progressively hard sampling strategy could be studied to enhance the robustness and effectiveness of our method. This will be added as limitations and future research in the revised version.
>
> - Figures:
>   We will enhance the color contrast, add more textual notes for figures. An additional figure of our framework will be added in the revised version.
>
> ---
>
> We believe these responses and revisions will address your concerns and those updated content will significantly strengthen our paper.
>
> Thank you again for your recognition of the novelty of our paper and invaluable feedback.

---

> > ### Comment · Reviewer_p7UF · 2025-08-05
> >
> > Thank you for your point-by-point response, which effectively addressed my concerns. I appreciate the clarity and effort in your rebuttal. I will take your explanations into account along with the comments from the other reviewers in making my final recommendation.

---

> ### Author Response · Authors · 2025-08-05
>
> **BMW achieves *SOTA* Performance on the Vehicle ReID Task on VeRi-776**
>
> ----
>
> Hi, we further evaluate our method on VeRi-776, which is a vehicle ReID dataset. Our BMW method also outperforms other methods on VeRi-776.
>
>
> We received the response from authors of the VeRi-776 dataset and got the data about 12 hours ago. We performed a quick validation of our BMW method on VeRi-776 and our BMW method also achieves SOTA performance on VeRi-776. Comparisons with other methods are given as follows:
>
> |Method|Reference|Source Data|mAP|
> |:-|:-:|:-:|:-:|
> MMT |ICLR 2020 | VehicleID | 35.3|
> |SpCL |NeurIPS 2020 | VehicleID| 38.9|
> |Cluster Contrast  |ACC 2022|-|40.3|
> |Cluster Contrast \[*Infomap*\] |ACC 2022|-|40.8|
> |DCC(ResNet50)| arxiv 2022 |-| 41.4 |
> |PPLR(*part features*)| CVPR 2022|-|41.6|
> |MLA |MTA 2024|-|38.4|
> |BMW|This paper|-|**42.1**|
> ||
> |DCC(*ResNet50-ibn*) |arxiv 2022 |-| 42.1 |
> |PPLR(*part features* + *camera labels*)| CVPR 2022|-|43.5|
> |PPLR(*part features* + *camera labels* + *re-ranking*)| CVPR 2024|-|45.3|
> ||
>
>
> It can be observed from the experiments on VeRi-776 that:
> - Our BMW method also outperforms other methods on the vehicle ReID task. For example, although PPLR(*part features*) uses extra part features, our BMW still outperforms it by 0.5% in mAP.
> - Our BMW method is even comparable with methods that use deeper backbone \[DCC(*ResNet50-ibn*)\], extra annotations \[PPLR (*part features* + *camera labels*)\] or stronger re-ranking \[PPLR(*part features* + *camera labels* + *re-ranking*)\].
> - As we use the same hyper-parameters on the VeRi-776 dataset, *i.e.*, $\lambda _ {intra}=0.9$ and $\lambda _ {inter}=0.2$, experiments on VeRi-776 further show that hyper-parameters selected on one dataset can be applied to other datasets with promising performance.
>
> Thus, we can conclude that: (1) our BMW method is effective for training with a memory bank and (2) hyper-parameter tuning is consistent for different datasets.
>
> ---
>
> We believe that experiments on VeRi-776 will help to validate the generality of our BMW method and also strengthen our paper.
>
> Thank you again for your constructive feedback!

---

> ### Author Response · Authors · 2025-08-05
>
> We sincerely appreciate your time and constructive feedback. We remain open to addressing any further concerns and would welcome continued discussion. Your insights are invaluable in improving the quality of our paper.

---

> > ### Comment · Reviewer_p7UF · 2025-08-06
> >
> > The proposed BMW framework introduces a novel memory updating mechanism by formulating it as a gradient descent optimization process, in contrast to the traditional momentum-based update. This perspective essentially treats the memory bank as a learnable classifier, where each memory slot is explicitly optimized to better represent class centers during training.
> >
> > Given this, I would like to ask a further question: Since the memory bank in BMW effectively functions as a learnable classifier, what are the key advantages of this design over using a standard learnable classifier such as a fully connected (FC) layer? Could the authors elaborate on the benefits of this approach in terms of handling pseudo-label instability, improving training robustness, enhancing feature representation, or providing better generalization, especially in the context of unsupervised person ReID?

---

> ### Author Response · Authors · 2025-08-06
>
> Thanks for your further feedback!
>
> It's an important question regarding the advantages of the memory bank over the FC layer in unsupervised person ReID. Below, we provide a detailed explanation:
>
> - **Memory bank enhances feature representation more effectively.**
>   1. **Fundamental difference between ReID and classification:** Unlike classification tasks, where training and test sets share the same categories, ReID involves disjoint identity sets between training and testing (because we cannot know identities that appear in the future). Thus, **a classifier trained on the training set cannot generalize to unseen identities in the test set**. Instead, the key challenge in ReID lies in learning discriminative feature representations where distance relationships (e.g., intra-class compactness and inter-class separation) are optimized, rather than classification accuracy.
>   2. **Limitations of FC classifiers in feature optimization:** A standard classifier (FC + SoftMax) minimizes cross-entropy loss to improve prediction accuracy but does not directly optimize feature relationships. As extensively discussed in prior works [R1-R9], this indirect optimization leads to suboptimal feature representations.
>   3. **Memory bank is more effective to optimize features:**
>   Unlike FC layers, the memory bank is optimized separately with the features. In our paper, the memory bank is optimized with $\mathcal{L} _ {\mathcal{M}}$ in Eqn.(4), while the features are optimized with $\mathcal{L} _ {\theta}$ in Eqn.(1).  **The target of memory bank update is to enhance the feature distance optimization and the performance of ReID, rather than accurately predicting image labels.**
>   This mechanism ensures that the learned features are more discriminative for ReID, rather than merely improving classification scores. For example, in the training phase on Market, although $\mathcal{L} _ {\theta}$ increases from 0.439 to 3.033 in the epoch 1, mAP on test set increases from 12.9% to 25.6% (training log will be provided on github).
> - **Memory bank is more robust to label noise induced by unsupervised clustering.**
>   1. **FC parameters are highly sensitive to label noise:** As the FC layer is trained via cross-entropy loss, a single noisy sample can distort all FC parameters through backpropagation, significantly degrading performance [R18-R21].
>   2. **Memory bank alleviates noise impact:**  In contrast, a noisy sample primarily affects only its corresponding memory bank. Other memory banks remain largely unaffected unless the noisy sample is selected as a hard negative, which is a rare occurrence given the large number of identities in ReID datasets. This inherent robustness makes the memory bank more stable under noisy unsupervised clustering.
>
> **Given its superior feature optimization and robustness to noise, the memory bank shows stronger generalization capacity in unsupervised ReID, as supported by prior studies [R10-R17].**
>
> ---
> Performance comparison between BMW and recent FC-based methods (MMT, MebNet, GCMT and PPLR):
>
> |Method|Reference|mAP on Market|mAP on Duke|mAP on MSMT17|
> |-|-|-|-|-|
> |MMT [mean teacher+ResNet50-IBN]|ICLR'20|76.6 [use Duke]|68.1 [use Market]|29.7 [use Duke]|
> |MebNet [mean teacher+DenseNet-121]|ECCV'20|76.0 [use Duke]|66.1 [use Market]|-|
> |GCMT [mean teacher]|IJCAI'21|79.7 [use Duke+CUHK03+MSMT-17]|69.1 [use Market+CUHK03+MSMT-17]|26.6 [use Duke]
> |PPLR [part features]|CVPR'22|84.4 [use camera label]|-|42.2 [use camera label]|
> |BMW|This paper|**86.3**|**75.2**|**44.6**|
> ||
>
> It's clear that, although FC-based methods use stronger backbones [ResNet50-IBN or DenseNet-121], extra annotations [other labeled dataset or camera label], or complicated structures [mean teachers or part features], our BMW method still outperforms them by a large margin. These comparison results further show that, memory banks are stronger for unsupervised person ReID.
>
> ---
> Reference:
>
> [R1] Deep learning ... verification. NeurIPS'14.
>
> [R2] Facenet. CVPR'15.
>
> [R3] Ring loss. CVPR'18.
>
> [R4] In defense of triplet ... arXiv'17.
>
> [R5] Point to set ... ReID. CVPR'17.
>
> [R6] Beyond triplet ... ReID.CVPR'17.
>
> [R7] Large margin ... ReID. T-MM'17.
>
> [R8] Group-group loss ... ReID. T-IP'19.
>
> [R9] Vehicle ReID ... aware metric learning. ICCV'19.
>
> [R10] Invariance matters ... ReID. CVPR'19.
>
> [R11] SpCL. NeurIPS'20.
>
> [R12] Cluster contrast ... ReID. ACCV'22.
>
> [R13] Dual Cluster ... ReID. arXiv'22.
>
> [R14] Discrepant and multi ... ReID. ICCV'23.
>
> [R15] Diverse semantic ... ReID. IF'24.
>
> [R16] Multi-level self ... ReID. MTA'24.
>
> [R17] Adapt only once ... ReID. PR'24.
>
> [R18] Unsupervised ReID ... soft multilabel. CVPR'19.
>
> [R19] Unsupervised ReID ... Multi-label. CVPR'20.
>
> [R20] MMT. ICLR'20
>
> [R21] GCMT. IJCAI'21.
>
> ---
> We believe our responses will address your concerns.
> We're here to address any remaining concerns and welcome further discussion. Thank you again for your invaluable feedback.

---

> > ### Comment · Reviewer_p7UF · 2025-08-07
> >
> > Thank you for your response. I will carefully consider how to improve my score. You mentioned that FC parameters are highly sensitive to label noise and that the memory bank alleviates the impact of noise. In this context, I wonder whether the proposed BMW is also effective under a supervised setting, offering more robust performance improvements compared to conventional classifiers. If so, I believe this could represent an even more significant contribution of the paper.

---

> ### Author Response · Authors · 2025-08-08
>
> Thanks for your constructive suggestion!
>
> On the supervised ReID task, memory bank can also enhance feature representations more effectively compared to conventional classifiers. Below, we provide the detailed explanation and validation.
>
> **1. Why is BMW more effective at feature optimization.**
>
> - Different from classification tasks, ReID requires learning discriminative feature representations where feature distance relationships (*e.g.*, intra-class compactness and inter-class separation) are optimized, rather than classification accuracy.
> - As conventional classifiers focus on accurate label prediction and do not directly optimize feature relationships, their abilities to optimize feature representations are limited.
> - Compared to conventional classifiers, the proposed BMW directly optimizes feature distance relationships with $\mathcal{L} _ {\theta}$ in Eqn.(1), and memory banks are optimized with $\mathcal{L} _ {\mathcal{M}}$ in Eqn.(4) to facilitate the feature representation learning.
>
> From the above analysis, the superiority of our BMW in feature representation learning over conventional classifiers stems from its ability to learn discriminative features through direct optimization of feature relationships, which is independent of data annotations. Consequently, although BMW is designed for unsupervised learning, this advantage naturally extends to supervised ReID tasks,
>
> **2. Experimental validation.**
>
> We further trained our BMW method under the supervised setting. The performance comparison between our BMW and conventional classifiers is summarized as follows:
>
> |Method|Type|Reference|mAP on Market|mAP on Duke|mAP on MSMT17|
> |-|-|-|:-:|:-:|:-:|
> |Only classifier|Classifier|IJCAI'21|81.1|70.4|-|
> |Classifier+circle loss|Classifier|CVPR'20|87.4|-|52.1|
> ||
> |ICE|Memory bank|ICCV'21|86.6|76.5|50.4|
> |ISE|Memory bank|CVPR'22|87.8|-|51.0|
> |DCMIP|Memory bank|ICCV'23|89.2|-|62.8|
> |DCC|Memory bank|arXiv'22|89.9|-|65.5|
> |**Unsupervised** BMW|Memory bank|This paper|86.3|75.2|44.6|
> |Supervised BMW|Memory bank|This paper|**90.5**|**80.2**|**66.7**|
> ||
>
> It can be observed from the above table that:
> - Compared to the conventional classifier, our BMW still enhances feature representations more effectively under the supervised setting.
> - Our unsupervised version of BMW is even comparable to the supervised circle loss method, which is a metric learning method. This shows the stronger feature optimization capacity of our BMW method.
> - Our BMW also outperforms other methods using memory bank by a clear margin. This further shows the superiority of our BMW method.
>
> ---
>
> The above explanation and performance comparison show that our BMW method is also more effective than conventional classifiers under the supervised setting. We believe these will further strengthen our paper.
>
> We're at your disposal and welcome further discussion. Thank you again for your time and constructive suggestion!

---

> > ### Comment · Reviewer_p7UF · 2025-08-09
> >
> > Thank you for your positive feedback. I will raise my score accordingly and look forward to the release of BMW as open source.

---

> > > ### Author Response · Authors · 2025-08-09
> > >
> > > We sincerely appreciate your time and constructive feedback. Your insights are invaluable in improving the quality of our paper.

---

### Official Review · Reviewer_RyMZ · 2025-07-02

**Clarity:** 3
**Significance:** 4
**Originality:** 4
**Rating:** 5
**Confidence:** 5

**Summary:**

This paper proposes a novel method for memory bank update, which is a core procedure in unsupervised learning of person re-identification models. Different from previous works, the proposed method introduces an inter-class constraint in the memory bank update as well as an intra-class constraint. Also, a weighting strategy for controlling the trade-off between these two contrainsts boosts the performance. Comprehensive empirical studies demonstrate the effectiveness of the proposed methods.

**Questions:**

Please see the Weaknesses.

**Ethical Concerns:**

["NO or VERY MINOR ethics concerns only"]

**Final Justification:**

This work proposes a novel unsupervised person re-identification method with sufficient contributions. I recommend to accept this paper.

**Limitations:**

Yes

**Paper Formatting Concerns:**

No problem

**Quality:**

3

**Strengths And Weaknesses:**

[Strengths]
- This paper is very well-written and easy to follow.
- The motivation for the inter-class constraint in memory bank update is clear and the analysis on the memory bank update mechanism from the gradient descent perspective is intriguing, providing useful insights.
- The proposed method is simple yet effective.
- Comprehensive experiment results demonstrating the proposed method are provided.

[Weaknesses]
- The experiments are conducted only with a ResNet50 backbone. Results with other backbones such as ViT would help validate the generality of the method.
- Some implementation details on how Equations (8) and (9) are implemented in practice (e.g. explanation with a mini-batch) are not clear. More clarification would be helpful.
- Minor typos are found: (1) In Equations (8) and (10), the term M[i] appears twice in $L_{inter}$, (2) In Figure 2’s legend, the red text includes a duplicated M[j], (3) Some numerical values in Table 3 appear to require scaling (e.g., 0.383 should likely be 38.3).

---

> ### Author Rebuttal · Authors · 2025-07-30
>
> **Response to the Reviewer RyMZ**
>
> ----
>
> We sincerely appreciate your recognition of the paper's novelty, originality, and clarity. And we thank you very much for your time and insightful comments. The point-by-point detailed response and the revision plans to each concern are given as follows:
>
> ---
> **Q1:** Experiments with other backbones.
>
> **A1:** Thank you for the valuable suggestion!
> Currently, ResNet50 is the most widely used backbone for unsupervised person ReID. Therefore, we adopt it in our paper ensuring fair comparisons with other approaches.
> To further validate our method’s generality, we conduct additional experiments with ResNet18, ResNet34, and ViT-B/16. Key results are summarized below:
>
> - ResNet18:
>    |Method|mAP on Market-1501|mAP on DukeMTMC-reID|
>    |:-:|:-:|:-:|
>    |w/o $\mathcal{L} _ {inter}$|57.1|52.2|
>    |BMW|**67.9**|**58.0**|
>    ||
>
> - ResNet34:
>    |Method|mAP on Market-1501|mAP on DukeMTMC-reID|
>    |:-:|:-:|:-:|
>    |w/o $\mathcal{L} _ {inter}$|65.4|60.8|
>    |BMW|**71.1**|**64.5**|
>    | |
>    |MMT[ICLR'20\] with ResNet50|71.2|63.1|
>    |IICS\[CVPR'21\] with ResNet50|72.9|64.4|
>    ||
>
> - ViT-B/16 (The initial weights are pretrained on the ImageNet-21K and then finetuned on ImageNet-1K following TransReID\[ICCV'21\]. We do not use the jigsaw patch shuffling module or side information embedding that are proposed in TransReID\[ICCV'21\]):
>    |Method|mAP on Market-1501|mAP on DukeMTMC-reID|
>    |:-:|:-:|:-:|
>    |w/o $\mathcal{L} _ {inter}$|84.6|73.5|
>    |BMW|**87.6**|**77.3**|
>    ||
>
> The mAPs yielded with ResNet18 and ResNet34 are lower than those with ResNet50 due to the smaller structures of the former two networks.
> The performances with ViT-B/16 are slightly better than that with ResNet50 because of its stronger capacity. Due to the limited time for those additional experiments, we do not adjust hyper-parameters with those backbones and only evaluate our core contribution, *i.e.*, the inter-class constraint.
>
> It's clear that, with different backbones, the proposed inter-class constraint can always boost the ReID performance.
> These experimental results show that our BMW method could generalize well with different backbone.
> This is because our method is proposed to update the memory bank with L2-normalized features and it is backbone-agnostic. Thus, our BMW can be applied to any backbones without modification.
>
> The experimental results demonstrate that the proposed inter-class constraint consistently enhances ReID performance across all tested backbones, confirming the strong generalizability of our BMW method. This adaptability stems from our backbone-agnostic design: the memory bank is updated with L2-normalized features. Therefore, BMW seamlessly integrates with any backbone architecture without requiring modifications.
>
> Notably, our method achieves superior performance even with smaller backbones.  For instance, our BMW with ResNet34 attains 64.5 in mAP on DukeMTMC-reID, outperforming IICS\[CVPR'21\] and MMT\[ICLR'20\] that use ResNet50. This result further demonstrates the effectiveness of our approach in ReID model training.
>
> Additionally, as detailed in our response to Reviewer a7FY, we have included supplementary experiments on DukeMTMC-reID using ResNet50. These results:
>
> - Show our method's clear performance advantage over competitors on DukeMTMC-reID.
>
> - Exhibit superior performance when applying the same hyperparameters across different datasets without additional tuning.
>
> While these DukeMTMC-reID results further validate our method's generalizability, data usage restrictions require us to include them only in our GitHub repository rather than the final paper.
>
> ---
> **Q2:** Details for Equations (8) and (9).
>
> **A2:** Thanks for pointing out that.
> In each training batch, we randomly sample 16 identities, *i.e.*, clusters by DBSCAN. For each sampled identity, 16 images are randomly selected.
> For inter-class constraint computation ($\mathcal{L} _ {inter}$), we update the 16 memory banks corresponding to those sampled 16 identities by their closest memory bank vector (hard sampling strategy) via $\mathcal{L} _ {inter}$.
> These details will be added in the revised version for a clearer description.
>
> By the way, we also provide detailed descriptions about the derivation process in our response to the Reviewer a7FY. We hope those will help to clarify our method.
>
>
> ---
> **Q3:** Textual errors and formatting issues.
>
> **A3:** We have carefully proofread the manuscript and fixed these issues:
>
> - Equations (8) and (10): We correct variable labels as:
> In Equation (8): $\mathcal{L} _ {inter}(\mathcal{M}[i],\mathcal{M}[j])=\frac{1}{2}\|\mathcal{M}[i]+\mathcal{M}[j]\| _ 2^2.$
> In Equation (10): $\mathcal{L}_{inter}(\mathcal{M}[i],\mathcal{M}[j])=\frac{1}{8}\|\mathcal{M}[i]+\mathcal{M}[j]\|_2^4.$
>
> - We enhance the color contrast and correct the legend for Figure 2. We also update the caption of Figure 2 as follows:
>
>      "Figure 2: Illustration of the advantage in inter-class constraint of the proposed objective function in Eqn.(8) w.r.t. different angles between $\mathcal{M}[i]$ and $\mathcal{M}[j]$. Yellow arrows and blue arrows denote  $\mathcal{M}[i]$ and $\mathcal{M}[j]$, respectively. Green dashed arrows and red dashed arrows denote the updated $\mathcal{M}[i]$ by the proposed objective function in Eqn.(8) and the previous method. It can be observed that the objective function in Eqn.(8) is more effective in pushing $\mathcal{M}[i]$ away from $\mathcal{M}[j]$."
>
>
> - Data format in Table 3: We unify the data in Table 3 to percentage format with 1 decimal place.
>
> These issues along with several typos will be corrected in the final version.
>
> ---
> We believe these revisions will address your concerns and significantly strengthen our paper.
>
> Thank you again for your recognition of our paper and invaluable feedback.

---

> ### Author Response · Authors · 2025-08-05
> **Validation on VeRi: Our BMW Achieves SOTA Performance**
>
> **BMW achieves *SOTA* Performance on the Vehicle ReID Task on VeRi-776**
>
> ----
>
> Hi, we further evaluate our method on VeRi-776, which is a vehicle ReID dataset. Our BMW method also outperforms other methods on VeRi-776.
>
>
> We received the response from authors of the VeRi-776 dataset and got the data about 12 hours ago. We performed a quick validation of our BMW method on VeRi-776 (which costs about 10.5 hours as VeRi-776 has more images) and our BMW method also achieves SOTA performance on VeRi-776. Comparisons with other methods are given as follows:
>
> |Method|Reference|Source Data|mAP|
> |:-|:-:|:-:|:-:|
> MMT |ICLR 2020 | VehicleID | 35.3|
> |SpCL |NeurIPS 2020 | VehicleID| 38.9|
> |Cluster Contrast  |ACC 2022|-|40.3|
> |Cluster Contrast \[*Infomap*\] |ACC 2022|-|40.8|
> |DCC(ResNet50)| arxiv 2022 |-| 41.4 |
> |PPLR(*part features*)| CVPR 2022|-|41.6|
> |MLA |MTA 2024|-|38.4|
> |BMW|This paper|-|**42.1**|
> ||
> |DCC(*ResNet50-ibn*) |arxiv 2022 |-| 42.1 |
> |PPLR(*part features* + *camera labels*)| CVPR 2022|-|43.5|
> |PPLR(*part features* + *camera labels* + *re-ranking*)| CVPR 2024|-|45.3|
> ||
>
>
> It can be observed from the experiments on VeRi-776 that:
> - Our BMW method also outperforms other methods on the vehicle ReID task. For example, although PPLR(*part features*) uses extra part features, our BMW still outperforms it by 0.5% in mAP.
> - Our BMW method is even comparable with methods that use deeper backbone \[DCC(*ResNet50-ibn*)\], extra annotations \[PPLR (*part features* + *camera labels*)\] or stronger re-ranking \[PPLR(*part features* + *camera labels* + *re-ranking*)\].
> - As we use the same hyper-parameters on the VeRi-776 dataset, *i.e.*, $\lambda _ {intra}=0.9$ and $\lambda _ {inter}=0.2$, experiments on VeRi-776 further show that hyper-parameters selected on one dataset can be applied to other datasets with promising performance.
>
> Thus, we can conclude that: (1) our BMW method is effective for training with a memory bank and (2) hyper-parameter tuning is consistent for different datasets.
>
> ---
>
> We believe that experiments on VeRi-776 will help to validate the generality of our BMW method and also strengthen our paper.
>
> Thank you again for your constructive feedback!

---

### Official Review · Reviewer_a7FY · 2025-07-05

**Clarity:** 1
**Significance:** 3
**Originality:** 3
**Rating:** 4
**Confidence:** 5

**Summary:**

This paper proposes a new memory bank rewriting mechanism called bidirectional memory bank rewriting (BMW), which represents memory bank rewriting as a gradient descent update with two objectives to improve category recognition and enhance feature discrimination capabilities. Several techniques are used to effectively improve memory bank representation capabilities for the two objectives and optimization, thereby achieving ReID feature optimization.

**Questions:**

1.This paper lacks validation across multiple datasets and different tasks, making it difficult to fully demonstrate the model's superiority.

2.The paper contains numerous textual errors and formatting issues.

3.The dynamic weighting scheme lacks detailed description, making the derivation process of Equations 10 to 11 difficult to understand.

4.The paper's charts and graphs are unclear, affecting readability and information comprehension.

**Ethical Concerns:**

["NO or VERY MINOR ethics concerns only"]

**Final Justification:**

The authors' clarification helps to resolve the main issues I previously raised.

**Limitations:**

yes

**Paper Formatting Concerns:**

Conforms to the Paper Formatting Instructions.

**Quality:**

2

**Strengths And Weaknesses:**

Strengths:

1.This paper proposes a bidirectional objective gradient descent update method for memory bank rewriting, which aims to improve feature discrimination capabilities.

2.For two-objective updates, this paper proposes an L2 norm and dynamic weighting strategy, which effectively improves memory bank representation capabilities and achieves ReID feature optimization.

Weaknesses:

1.The model was not evaluated on the VeRi-776 and DukeMTMC-reID datasets, limiting the ability to validate its overall superiority. Furthermore, the results in Table 1 indicate suboptimal performance on the MSMT-17 dataset.

2.Several minor errors are present in the manuscript, including incorrect labeling of variables i and j in Equations (8) and (10).

3.Figures 1 and 2 lack clarity, making it difficult for readers to understand the presented concepts.

4.The data format in Table 1 is inconsistent, mixing percentages and decimal values. Additionally, the proposed method is highly sensitive to hyperparameters and lacks cross-dataset hyperparameter analysis, limiting its applicability to real-world recognition tasks.

5.The introduction lacks analytical rigor, the literature review is insufficient, and the inter-class constrained memory bank approach is not original to this work. Moreover, although domain adaptation for pedestrian re-identification is mentioned, the experimental section does not include relevant analysis.

---

> ### Author Rebuttal · Authors · 2025-07-30
>
> **Response to the Reviewer a7FY**
>
> ---
> Thank you very much for your time and insightful comments. We sincerely appreciate your recognition of our core contribution. Point-by-point responses and revision plans to each concern are given as follow:
>
> ---
> **Q1:** Validation on DukeMTMC-reID and VeRi-776, performance on MSMT-17 in Table 1 (actually it's Table 3 in our manuscript), and cross-dataset hyperparameter analysis.
>
> **A1:** We provide the following clarification for each dataset:
>
> - DukeMTMC-reID (Duke for short) and cross-dataset hyperparameter analysis:
> Actually, we have compared our methods on Duke with other approaches before the submission. However, Duke has been taken down while some previous conferences, e.g., ICCV 2025, explicitly prohibited using it. Due to the concerns of this restriction, we removed the results on Duke from the comparison with SOTAs in Table 3. We provide those results as:
>   - Comparison with recent SOTAs. Note that works after 2022 commonly don't use Duke.
>
>     |Method|Reference|Source Data|mAP|Rank1|Rank5|Rank10|
>     |-|-|:-:|:-:|:-:|:-:|:-:|
>     |MMT|ICLR'20|Market-1501|63.1|76.8|88.0|92.2|
>     |MEB-Net|ECCV'20|Market-1501|66.1|79.6|88.3|92.2|
>     |CCL+PDA+FA|ICCV'21|Market-1501|70.8|83.5|-|-|
>     |GCMT|IJCAI'21|-|63.6|78.2|88.6|91.3|
>     |GCMT|IJCAI'21|Market-1501|67.8|81.1|91.1|93.9|
>     |IICS|CVPR'21|-|64.4|80.0|89.0|91.6|
>     |P $^2$ LR|AAAI'22|Market-1501|70.8|82.6|90.8|93.7|
>     |IIDS|T-PAMI'24|-|68.7|82.1|90.8|93.7|
>     |NPSSL|T-ETCI'25|-|67.6|78.9|88.6|92.0|
>     |BMW|This paper|-|**75.2**|**86.0**|**92.6**|**95.0**|
>     ||
>
> The above comparison on Duke shows that our BMW method also achieves the best performance. For instance, although P $^2$ LR \[AAAI'22\] uses extra training data, BMW still outperforms it by 4.2% in mAP. This further demonstrates the effectiveness of our BMW.
>
> We further conduct the parameter analysis on Duke as follows:
>   - Evaluation on $\lambda_{intra}$
>     ||||||||
>     |-|-|-|-|-|-|-|
>     |$\lambda_{intra}$|0.5|0.6|0.7|0.8|0.9|1|
>     |mAP on Duke|69.7|73.6|74.5|75.0|**75.2**|75.1|
>     ||
>   - Evaluation on $\lambda_{inter}$
>     |||||||||
>     |-|-|-|-|-|-|-|-|
>     |$\lambda_{inter}$|0|0.05|0.1|0.15|0.2|0.25|0.3|
>     |mAP on Duke|67.8|70.6|72.5|74.4|**75.2**|74.1|65.5|
>     ||
>   - Evaluation on $\tau$
>     |||||||
>     |-|-|-|-|-|-|
>     |$\tau$|0.01|0.03|0.05|0.07|0.1|
>     |mAP on Duke|74.4|75.0|**75.2**|73.5|72.1|
>     ||
>   - Ablation study of BMW
>
>     |||||||
>     |-|:-:|:-:|:-:|:-:|-|
>     |Method|baseline [15]|w/o $\mathcal{L}_{inter}$|$\|\|\mathcal{M}[i]-\mathcal{M}[j]\|\|_2^2$ in Eqn.8|w/o dynamically weighting|BMW|
>     |mAP on Duke|66.5|67.8|71.6|73.3|**75.2**|
>     ||
>
> The model analysis results on Duke show that the best performance is achieved when $\lambda_{intra}=0.9$, $\lambda_{inter}=0.2$, and $\tau=0.05$, which are the optimal parameters for BMW on Market-1501. This indicates that hyperparameters selected on one dataset can be applied to others without sacrificing the performance. Meanwhile, the superior performance does not totally rely on careful selection of hyperparameters. We notice that BMW consistently achieves promising performance as long as $\lambda_{intra}$ sits in $[0.6,1]$ and $\lambda_{inter}$ in $[0.1,0.25]$.
>
> The superiority range of $\lambda_{inter}$ is smaller than that of $\lambda _ {intra}$. This is because $\mathcal{L} _ {intra}$ decreases after several training iterations even when using the hard sampling strategy, while $\mathcal{L} _ {inter}$ stays large because of the large number of identities. We can apply a threshold on $\mathcal{L} _ {inter}$ to ignore memory bank pairs that are already far away from each other. This strategy will enlarge the operating range of $\lambda_{inter}$.
>
> The temperature hyperparameter $\tau$ is widely used in constrastive learning methods. It's commonly set to 0.05 in previous works and our BMW achieves promising performance when setting it in the range of \[0.01,0.05\].
>
> The ablation study on Duke shows that proposed methods effectively boost the ReID performance on differnet datasets.
>
> Due to the data usage restriction, we plan to include these results on Duke only in our github repository, not in the paper.
>
> - VeRi:
> Thank your for pointing out that VeRi can be used for validation on the vehicle ReID task. Currently, we are still awaiting the authors' response regarding data access. Experiments will be performed once we get the data.
>
> - MSMT-17:
> In Table 3, the results on MSMT-17 of MEB-Net and MMT are mistakenly listed as their performance on Duke. Due to the last-minute removal of Duke results in Table 3, we mistakenly left these entries showing Duke results. We correct these values as follows. Note that, MEB-Net does not use MSMT-17, it only reports its performance on Matker-1501 and Duke.
>
>     |Method|Reference|Source Data|mAP|Rank1|Rank5|Rank10|
>     |:-:|:-:|:-:|:-:|:-:|:-:|:-:|
>     |MMT|ICLR'20|Market-1501|26.6|54.4|67.6|72.9|
>     |MMT|ICLR'20|Duke|29.7|58.8|71.1|76.1|
>     |BMW|This paper|-|**44.6**|**75.5**|**86.3**|**87.1**|
>     ||
>
> From the corrected values, our BMW method also outperforms recent works by a clear margin on MSMT-17.
>
> Due to the large number of identities and the large variation in intra-ID images, MSMT-17 is more challanging than Market-1501 and Duke, thus current methods achieve lower performance on MSMT-17 than on Market-1501 and Duke.
>
> We will correct these entries in Table 3 and further clarify the comparison on MSMT-17 in the final version.
>
> ---
> **Q2:** Textual errors and formatting issues.
>
> **A2:** Thanks for pointing out that. We have carefully proofread and fixed these issues:
>
> - Equations (8) and (10): We correct variable labels as:
> In Equation (8): $\mathcal{L} _ {inter}(\mathcal{M}[i],\mathcal{M}[j])=\frac{1}{2}\|\mathcal{M}[i]+\mathcal{M}[j]\| _ 2^2.$
> In Equation (10): $\mathcal{L}_{inter}(\mathcal{M}[i],\mathcal{M}[j])=\frac{1}{8}\|\mathcal{M}[i]+\mathcal{M}[j]\|_2^4.$
>
> - Data format in Table 3: We unify the data in Table 3 to percentage format with 1 decimal place.
>
> These issues along with several typos will be corrected in the final version. We further group methods by their backbones and source data in Table 3 for easier comparison.
>
> ---
> **Q3:** Detailed description of the dynamic weighting scheme.
>
> **A3:** Good suggestion! The dynamic weight scheme is based on the idea that larger value of objective function needs larger weight. Hence, the weights for Eqn.(6) and Eqn.(8) are $\frac{1}{4}\|\mathcal{M}[c]-f_i\| _ 2^2$ and $\frac{1}{4}\|\mathcal{M}[i]+\mathcal{M}[j]\| _ 2^2$, respectively. Then, $\mathcal{L} _ {intra}$ and $\mathcal{L} _ {inter}$ are:
> - $\mathcal{L} _ {intra}(\mathcal{M}[c],f _ i)=\frac{1}{4}\|\mathcal{M}[c]-f _ i\| _ 2^2\times\frac{1}{2}\|\mathcal{M}[c]-f_i\| _ 2^2=\frac{1}{8}\|\mathcal{M}[c]-f _ i\| _ 2^4$,
>
> - $\mathcal{L} _ {inter}(\mathcal{M}[i],\mathcal{M}[j])=\frac{1}{4}\|\mathcal{M}[i]+\mathcal{M}[j]\| _ 2^2\times\frac{1}{2}\|\mathcal{M}[i]+\mathcal{M}[j]\| _ 2^2=\frac{1}{8}\|\mathcal{M}[i]+\mathcal{M}[j]\| _ 2^4.$
>
> Their gradient w.r.t. $\mathcal{M}[c]$ and $\mathcal{M}[i]$ are:
>
> - $\frac{\partial \mathcal{L}_{intra}(\mathcal{M}[c],f_i)}{\partial\mathcal{M}[c]}$
> $=\frac{1}{8}\times4\times\|\mathcal{M}[c]-f_i\|_2^3\times\frac{\mathcal{M}[c]-f_i}{\|\mathcal{M}[c]-f_i\|_2}$
> $=\frac{1}{2}\|\mathcal{M}[c]-f_i\|_2^2(\mathcal{M}[c]-f_i)$
> $=\frac{1}{2}(\|\mathcal{M}[c]\|_2^2+\|f_i\|_2^2-2(\mathcal{M}[c])^\top f_i)(\mathcal{M}[c]-f_i)$
> $=\frac{1}{2}(1+1-2(\mathcal{M}[c])^\top f_i)(\mathcal{M}[c]-f_i)$
> $=(1-(\mathcal{M}[c])^\top f_i)(\mathcal{M}[c]-f_i)$,
>
> - $\frac{\partial \mathcal{L}_{inter}(\mathcal{M}[i],\mathcal{M}[j])}{\partial\mathcal{M}[i]}$
> $=\frac{1}{8}\times4\times\|\mathcal{M}[i]+\mathcal{M}[j]\|_2^3\times\frac{\mathcal{M}[i]+\mathcal{M}[j]}{\|\mathcal{M}[i]+\mathcal{M}[j]\|_2}$
> $=\frac{1}{2}\|\mathcal{M}[i]+\mathcal{M}[j]\|_2^2(\mathcal{M}[i]+\mathcal{M}[j])$
> $=\frac{1}{2}(\|\mathcal{M}[i]\|_2^2+\|\mathcal{M}[j]\|_2^2+2(\mathcal{M}[i])^\top \mathcal{M}[j])(\mathcal{M}[i]+\mathcal{M}[j])$
> $=\frac{1}{2}(1+1+2(\mathcal{M}[i])^\top \mathcal{M}[j])(\mathcal{M}[i]+\mathcal{M}[j])$
> $=(1+(\mathcal{M}[i])^\top \mathcal{M}[j])(\mathcal{M}[i]+\mathcal{M}[j])$.
>
> These will be added in the revised version for an easier understanding.
>
> ---
> **Q4:** Charts and graphs are unclear.
>
> **A4:** We will enhance the color contrast and add more textual notes for figures. Detailed captions for Figures 1 and 2 will be provided as follows:
>
>   - "Figure 1: Illustration of the memory bank update without (w/o) and with (w/) the inter-class constraint. Dots and stars denote features and memory banks. Different colors denote different clusters. Without the inter-class constraint, memory banks are pulled to approach positive samples only (shown as yellow arrows). With the inter-class constraint, memory banks are also push away from each other (shown as purple arrows), thus enhancing their separability."
>   - "Figure 2: Illustration of the advantage in inter-class constraint of the proposed objective function in Eqn.(8) w.r.t. different angles between $\mathcal{M}[i]$ and $\mathcal{M}[j]$. Yellow arrows and blue arrows denote $\mathcal{M}[i]$ and $\mathcal{M}[j]$, respectively. Green dashed arrows and red dashed arrows denote the updated $\mathcal{M}[i]$ by the proposed objective function in Eqn.(8) and the previous method, respectively. It can be observed that the objective function in Eqn.(8) is more effective in pushing $\mathcal{M}[i]$ away from $\mathcal{M}[j]$."
>
> These will be added in the revised version for clear illustrations.
>
> **Q5:** More related works and analysis in experiments.
>
> **A5**: Thanks for the advise. The final version will include (1) more related works, e.g., FUReID, IIDS, etc. (2) Analysis with domain adaptation ReID in experiments.
>
> ---
> We believe these revisions will address the concerns and significantly strengthen our paper.
>
> Thank you again for your invaluable feedback.

---

> ### Author Response · Authors · 2025-08-05
> **Validation on VeRi: Our BMW Achieves SOTA Performance**
>
> **BMW achieves *SOTA* Performance on the Vehicle ReID Task on VeRi-776**
>
> ----
>
> Following your suggestion of validation on VeRi-776, we further evaluate our method on the vehicle ReID application and our method also outperforms other methods.
>
> We received the response from authors of the VeRi-776 dataset and got the data about 12 hours ago. We performed a quick validation of our method on VeRi-776 (which costs about 10.5 hours as VeRi-776 has more images) and our BMW method also achieves SOTA performance on VeRi-776. Comparison with other methods are given as follows:
>
> |Method|Reference|Source Data|mAP|
> |:-|:-:|:-:|:-:|
> MMT |ICLR 2020 | VehicleID | 35.3|
> |SpCL |NeurIPS 2020 | VehicleID| 38.9|
> |Cluster Contrast  |ACC 2022|-|40.3|
> |Cluster Contrast \[*Infomap*\] |ACC 2022|-|40.8|
> |DCC(ResNet50)| arxiv 2022 |-| 41.4 |
> |PPLR(*part features*)| CVPR 2022|-|41.6|
> |MLA |MTA 2024|-|38.4|
> |BMW|This paper|-|**42.1**|
> ||
> |DCC(*ResNet50-ibn*) |arxiv 2022 |-| 42.1 |
> |PPLR(*part features* + *camera labels*)| CVPR 2022|-|43.5|
> |PPLR(*part features* + *camera labels* + *re-ranking*)| CVPR 2024|-|45.3|
> ||
>
>
> It can be observed from the experiments on VeRi-776 that:
> - Our BMW method also outperforms other methods on the vehicle ReID task. For example, PPLR(*part features*) uses extra part features and our BMW still outperforms it by 0.5% in mAP.
> - Our BMW method is even comparable with methods that use deeper backbone \[DCC(*ResNet50-ibn*)\], extra annotations \[PPLR (*part features* + *camera labels*)\] or stronger re-ranking \[PPLR(*part features* + *camera labels* + *re-ranking*)\].
> - As we use the same hyper-parameters on the VeRi-776 dataset, *i.e.*, $\lambda _ {intra}=0.9$ and $\lambda _ {inter}=0.2$, experiments on VeRi-776 further show that hyper-parameters selected on one dataset can be applied on other datasets with promising performance.
>
> Thus, we can conclude that: (1) our BMW method is effective in model training with the memory bank and (2) hyper-parameter tuning is consistent for different datasets.
>
> ---
>
> We believe that experiments on VeRi-776 will further address your concerns about the superiority of our method and will also significantly strengthen our paper.
>
> Thank you again for your constructive feedback!

---

> > ### Comment · Reviewer_a7FY · 2025-08-05
> >
> > The authors have provided satisfactory clarification and supporting evidence. My concerns have been addressed, and I will keep my score at 4.

---

> > > ### Author Response · Authors · 2025-08-05
> > >
> > > We sincerely appreciate your insightful comments and constructive feedback, which have greatly improved our paper. We are grateful for your time and efforts in reviewing our work.

---

### Note · Authors · 2025-08-14

We sincerely appreciate the AC and all Reviewers for their constructive feedback and for acknowledging the merits of our work.

This paper introduces the Bidirectionally Memory bank reWriting (BMW) method, which is innovatively formulated as a gradient descent update with novel intra-class and inter-class constraints.

In the rebuttal session, we have provided point-by-point detailed responses and additional experiments that not only address every raised concern but also further **highlight the advantages and generalizability** of our BMW method.

**Key Improvements**:

- **Extended Experiments.**
We have:
    1. Added results on **two large-scale datasets** (VeRi-776 and DukeMTMC-reID) demonstrating BMW’s consistent superiority over other SOTA methods.
    2. Verified the generalizability across **three backbone architectures** (ResNet18, ResNet34 and ViT-B/16).
    3. Conducted comparisons showing BMW's significant **advantages over FC classifiers** for ReID tasks.
    4. Performed **hyper-parameter analysis** confirming robustness of BMW across different datasets without hyper-parameter tuning.

- **Enhanced explanations.**
We have significantly strengthened the paper's analytical depth through:
  1. Rigorous analysis of **hyper-parameter impacts** on model performance.
  2. Detailed exposition of **constraint computation** in mini-batch training.
  3. Qualitative reasoning explaining BMW's **superior ability on feature learning**.
  4. Efficiency analysis demonstrating extra **minimal overhead in computation**.

- **Polished presentation:**
We have improved the paper's readability by refining the equations, figures, and notations.

These validations, explanations and clarifications significantly strengthen this paper and will be added to the final version.

Code will be released once accepted.

---

### Decision · Program_Chairs · 2025-09-17

**Decision:**

Accept (poster)

**Comment:**

This paper proposes BMW, a novel bidirectional memory bank rewriting mechanism for unsupervised person Re-ID that incorporates both intra-class and inter-class constraints through a gradient descent update framework. All four reviewers consider the paper's technical soundness and novelty. Reviewers RyMZ and p7UF strongly recommended acceptance, with clear motivation and comprehensive experiments. Reviewers a7FY and dA3y initially had concerns about dataset coverage, implementation details, and hyperparameter sensitivity, but both upgraded their scores to borderline accept after the authors provided convincing responses.